# Generative modeling of single-cell time series with PRESCIENT enables prediction of cell trajectories with interventions

Grace Hui Ting Yeo [1,2,5], Sachit D. Saksena [1,2,5] & David K. Gifford [1,3,4 ✉]

Existing computational methods that use single-cell RNA-sequencing (scRNA-seq) for cell fate prediction do not model how cells evolve stochastically and in physical time, nor can they predict how differentiation trajectories are altered by proposed interventions. We introduce PRESCIENT (Potential eneRgy undErlying Single Cell gradIENTs), a generative modeling framework that learns an underlying differentiation landscape from time-series scRNA-seq data. We validate PRESCIENT on an experimental lineage tracing dataset, where we show that PRESCIENT is able to predict the fate biases of progenitor cells in hematopoiesis when accounting for cell proliferation, improving upon the best-performing existing method. We demonstrate how PRESCIENT can simulate trajectories for perturbed cells, recovering the expected effects of known modulators of cell fate in hematopoiesis and pancreatic β cell differentiation. PRESCIENT is able to accommodate complex perturbations of multiple genes, at different time points and from different starting cell populations, and is available at https://github.com/gifford-lab/prescient.

[1] Computer Science and Artificial Intelligence Laboratory, Massachusetts Institute of Technology, Cambridge, MA, USA. [2] Computational and Systems Biology Program, Massachusetts Institute of Technology, Cambridge, MA, USA. [3] Department of Electrical Engineering and Computer Science, Massachusetts Institute of Technology, Cambridge, MA, USA. [4] Department of Biological Engineering, Massachusetts Institute of Technology, Cambridge, MA, USA. [5] These authors contributed equally: Grace Hui Ting Yeo, Sachit D. Saksena. ✉email: gifford@mit.edu

Modeling developmental landscapes is essential to improving our understanding of how cells are driven to transient and terminal states in vivo and to enable precise manipulation of cell fates in vitro[1]. Single-cell RNA-sequencing (scRNA-seq) has enabled the study of developmental landscapes by the observation of gene expression in single cells sampled at multiple stages of differentiation. However, these studies provide snapshots of a given differentiation process and do not directly observe lineage relationships between cells at different time points in development. Recently, experimental lineage tracing methods that couple various barcoding strategies with scRNA-seq have been described that identify lineage relationships[2–4]. These methods provide ground truth for computational models of differentiation.

Existing computational approaches for modeling differentiation typically summarize observations of cell states and couplings emergent of the underlying process and have limited to no capacity for modeling differentiation as a continuous process (Fig. 1a). The predominant approach is pseudo-temporal inference, which orders cells along an arbitrary one-dimensional measurement representing 'differentiation time', and hence cannot model differentiation dynamics with respect to real, physical time[5–7]. Other methods have also emerged for the specific task of cell fate prediction. For example, Waddington-OT predicts long-range cell–cell probabilistic couplings by reframing the task of inferring cell relationships between population snapshots as an unbalanced optimal transport problem[8]. Another method, Fate-ID iteratively builds ensembled cell-type classifiers from labeled terminal cell states[9]. However, these methods only summarize observations of cell states and couplings emergent of the underlying differentiation process. Recently, a small number of methods have described approaches to modeling differentiation as a process, but they have been limited either in how the model is solved, or in modeling capacity. For example, Population Balance Analysis (PBA) solves a reaction-diffusion partial differential equation describing differentiation but is forced to use a nonparametric solution due to computational constraints[10]. Similarly, pseudodynamics models a diffusion process but only in a one-dimensional cell state[11].

We introduce PRESCIENT (Potential eneRgy undErlying Single Cell gradIENTs), a generative modeling framework fit using longitudinal scRNA-seq datasets to model complex potential landscapes. PRESCIENT extends previous work by Hashimoto et al.[12] that showed that a global potential function of a time-series is recoverable via a diffusion-based model fit to well-mixed, cross-sectional observations. PRESCIENT builds upon this by enabling the model to operate on large numbers of cells over many timepoints with high-dimensional features, and by incorporating cellular growth estimates. We validate PRESCIENT on a newly published lineage tracing dataset by evaluating PRESCIENT's ability to generate held-out timepoints and to predict cell fate bias, i.e. the probability a cell enters a particular fate given its initial state. We show that when accounting for cell proliferation, PRESCIENT outperforms existing methods on predicting cell fate bias. Unlike existing methods, PRESCIENT learns a stochastic, parametric, queryable form of the differentiation landscape via a generative neural network, which enables simulations of high-dimensional trajectories with arbitrary initializations in physical time. This enables simulation of trajectories for cells unobserved during training, including cells with computationally perturbed gene expression profiles, which none of the existing summarization methods or modeling methods are able to do (Fig. 1a). This is also in contrast to other generative methods like scGen, which was proposed for predicting shifts in gene expression space in response to perturbations via autoencoder latent space arithmetic[13]. While scGen is a promising approach for generating cell profiles under different perturbations for initializations of PRESCIENT models (Discussion), it is not time resolved, does not generate distributions of cells, and does not explicitly model cellular differentiation. We demonstrate how PRESCIENT can be used to model perturbations of multiple genes, at different time points, and from different starting states. We are able to recover expected changes in final cell fate distributions when interrogating our models using perturbations of known regulators of cell fate in hematopoiesis and pancreatic β cell differentiation. PRESCIENT enables large unbiased in silico perturbation experiments to aid the design of in vitro genetic perturbational screens.

## Results

**Learning a generative model of cellular differentiation from high-dimensional scRNA-seq data**. PRESCIENT models cellular differentiation as a diffusion process over a gene expression landscape parameterized by a potential function that we wish to identify given only time-series population snapshots of single-cell RNA expression. In this diffusion process, evolution of a cell's state at a given time is governed by a drift term, corresponding to the force acting on that cell given its current state, and a noise term, corresponding to stochasticity. In particular, the drift term is defined to be the negative gradient of the potential function, such that the potential induces a force that naturally drives cells toward regions of low potential (Fig. 1b; Methods). This stochastic process can then be simulated via first-order time discretization to sample trajectories for a given cell. The potential function is fit by minimizing a regularized Wasserstein loss between empirical and predicted populations at the observed time points (Fig. 1c; Methods). Previous work has shown the recoverability of diffusion dynamics from cross-sectional observations via this objective function[12].

To enable the modeling framework to operate on large scRNA-seq datasets, we fit models on PCA projections of the scaled gene expression data, which has been successfully used in down-stream scRNA-seq analysis methods such as clustering and cross-dataset integration[14–16]. The potential function is given by a neural network, which operates as a black-box function approximator, hence enabling complex parameterizations of the landscape (Fig. 1c).

Finally, we take into account cell proliferation by weighting each cell in the source population according to its expected number of descendants in the objective. To assess the importance of incorporating cell proliferation, we study models assuming a priori knowledge of cell proliferation, which can be directly estimated from the data by computing the number of descendants for each starting cell given lineage tracing data, as well as models where cell proliferation is estimated from gene expression[8].

**Fate outcomes generated by PRESCIENT align with experimental lineage tracing when taking into account cell proliferation**. We validate our model on a recently published lineage tracing dataset by Weinreb et al. which used DNA barcodes to track clonal trajectories during mouse hematopoiesis[3]. We evaluate our model on two tasks: recovery of a held-out time point, and cell fate prediction (Fig. 1d, e).

We first evaluated whether models were able to recover the marginal cell population at a held-out time point, day 4, when trained only on days 2 and 6, using cells that have lineage tracing data available. We evaluated the Wasserstein distance between the simulated and the empirically observed cell populations for days 4 (testing distance) and 6 (training distance) on the epoch with the lowest training distance. Simulated populations generated by our model outperform baselines, including the distance of the

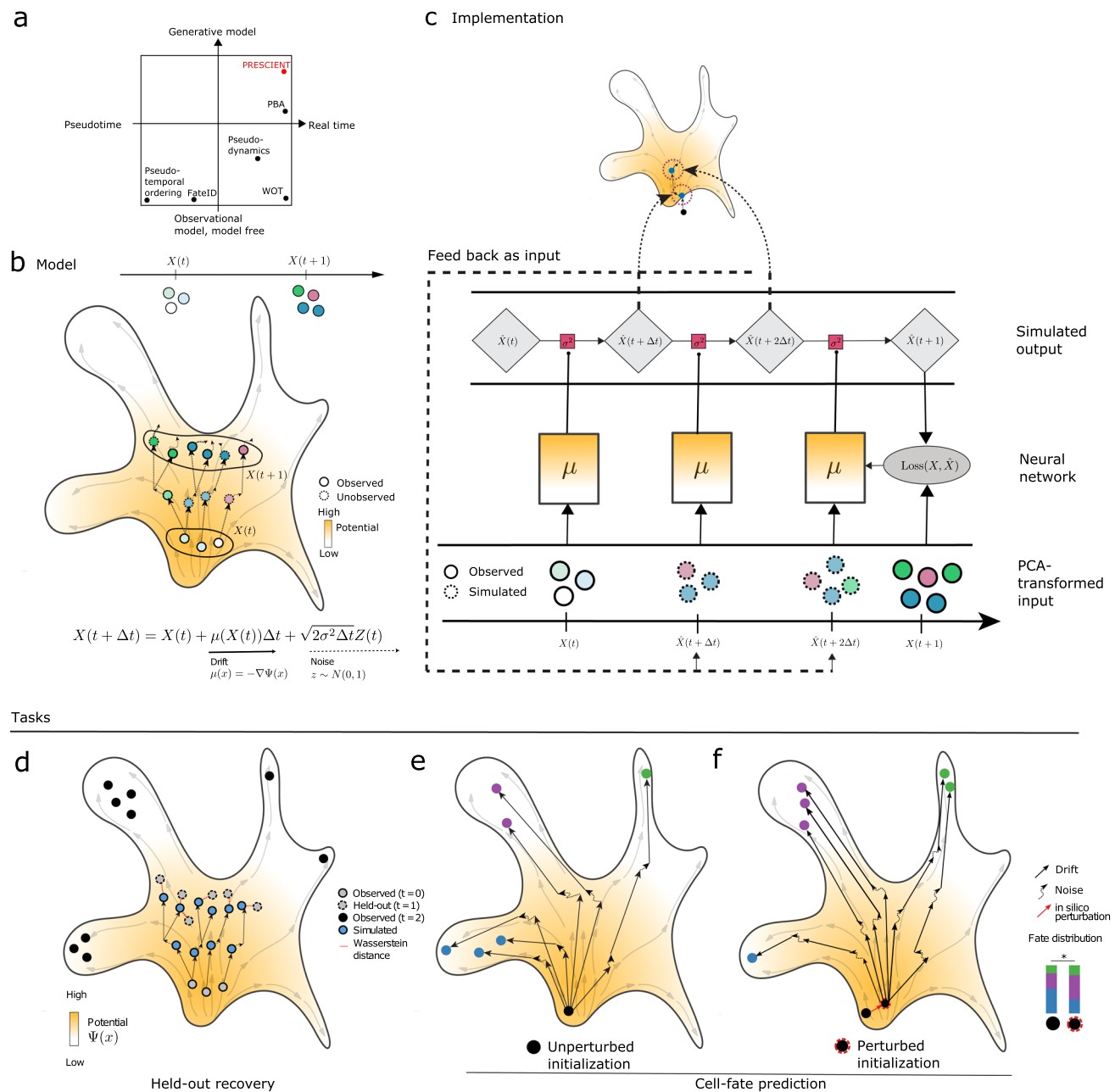

**Fig. 1 A generative model of cellular differentiation. a** Existing single-cell models of development can be described as operating in pseudo-time or real time (x-axis), and by the extent to which they model the underlying differentiation process (y-axis). PRESCIENT is highlighted in red. **b** Observations of population-level time-series data are used in a generative framework that models the underlying dynamic process in physical time. Evolution of a cell's state is governed by a drift term and a noise term. The drift, depicted by solid arrows, is defined as the negative gradient of the potential function, depicted by the color gradient in the background. Dashed lines correspond to noise. The model is fit using observations of population-level time-series data, depicted as solid circles. Simulations of cell states are depicted as dashed circles. **c** Cartoon depicting model fitting process. The neural network parameterizing the underlying drift function μ takes as input the PCA projections of gene expression data at observed time points (again depicted as solid lines). The stochastic process is then simulated via first-order time discretization to produce a population at the next time step, and so on. This proceeds until the next observed time point, at which the loss between the simulated and predicted population is minimized. The model was validated using two tasks. **d** Held-out recovery, where the model was asked to predict the marginal distribution of a held-out time point, and **e-f**, fate prediction, where the model was asked to predict the fate distribution outcome of a given progenitor cell. Fate prediction can be applied to cells observed in the dataset (**e**) or cell states in which some perturbation has been applied in silico (**f**). As shown, the perturbation results in a significant shift of fate distribution outcomes.

simulated population at day 4 to the actual populations at day 2 and 6, as well as a linearly interpolated population as predicted by WOT[8] (Fig. 2a, Supplementary Note).

We next evaluated on cell fate prediction, which we define to be the task of predicting the clonal fate bias of a given barcoded clone as described by Weinreb et al. This is defined as the number

of neutrophils divided by the total number of neutrophils and monocytes for that clone (Methods). To predict, we simulate 2000 trajectories initialized with only the starting cell of each clone until the final time point. For each of these trajectories, we classify the cell at the final time point as neutrophil, monocyte or other using an approximate nearest neighbor (ANN) classifier that had

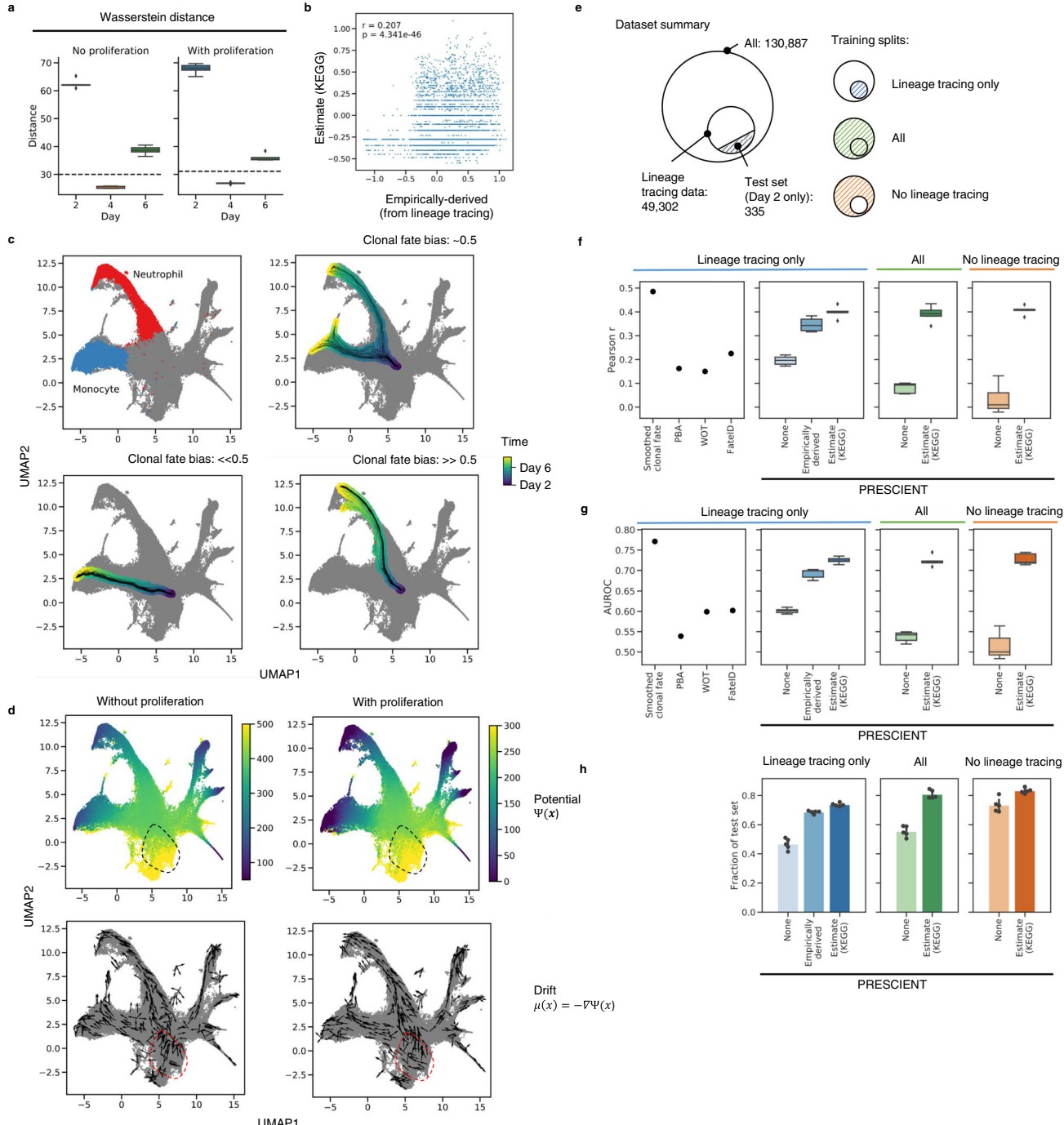

been trained using cell-type labels provided by Weinreb et al. (Methods). We then evaluate the clonal fate bias on the test set, by ensembling predictions over the last 5 evaluated epochs (Figs. 2c, d, S2a). We measure performance as the Pearson correlation with respect to the actual clonal fate bias given by the lineage tracing data, as well as AUROC of classifying a given cell as having a clonal fate bias of >0.5, since the metric is strongly bimodal (Fig. S2b).

We first fit a PRESCIENT model on the subset of data for which lineage tracing data is available (Fig. 2e). Figure 2d depicts the learned potential and drift functions for models trained with and without cell proliferation. Qualitatively, we observe that incorporating cell proliferation changes the potential landscape near the earliest time point. When the model does not take into account cell proliferation, its performance is similar to existing fate

prediction methods like Waddington-OT with provided empirically derived cell proliferation ($r = 0.150$, AUROC = 0.599) and FateID ($r = 0.225$, AUROC = 0.602), achieving $r = 0.196 \pm 0.020$, AUROC = $0.601 \pm 0.006$ over 5 seeds. Incorporating empirically derived cell proliferation rates into the PRESCIENT modeling framework greatly improves performance with mean $r = 0.347 \pm 0.029$, AUROC = $0.692 \pm 0.012$, more closely approaching the upper-bound performance estimated using held-out clonal data of $r = 0.487$, AUROC = 0.771 (Figs. 2f, g; S2).

Accounting for cell proliferation improves the performance of the fate prediction task. However, these models were fit with empirically derived cell proliferation rates calculated from the lineage tracing data, which is usually unavailable (Figs. 2e, S2c). We next looked at whether cell proliferation rates derived from gene expression could achieve similar performance. To this end,

**Fig. 2 Fate outcomes generated by PRESCIENT align with experimental lineage tracing when taking into account cell proliferation. a** Testing performance of models trained with and without proliferation on the task of recovering the held-out time point, day 4. Performance is reported as Wasserstein distance of simulated population with respect to actual populations at the given time points for models trained without and with proliferation over $n = 5$ seeds. Dashed line indicates distance of linearly interpolated population with respect to the actual population on day 4. **b** Correlation of actual and estimated proliferation rates on day 2 cells with lineage tracing data ($n = 4638$). **c** Examples of $n = 50$ trajectories with starting cells assigned different clonal fate biases. The color-scale indicates time corresponding to 100 time steps of $dt = 0.1$ starting from $t = 2$. Clonal fate bias is computed with respect to monocyte/neutrophil populations at the final time point. **d** Visualizations of underlying potential and drift functions learned by models with and without cell proliferation. Drift is visualized for a random sample of cells. Dotted circle indicates qualitative differences in potential landscape. **e** Summary of training/test splits of lineage tracing dataset. Training sets either include only cells with lineage tracing data, all cells, or cells without lineage tracing data. **f**–**h**, Performance of other methods (far left, Smoothed fate probabilities given by held-out clonal data, PBA: population balance analysis[10], WOT: Waddington-OT[8], FateID[9] evaluated using predictions provided by Weinreb et al.) in comparison to PRESCIENT (left-right) on predicting clonal fate bias for the same test set ($n = 335$) given different training sets. Training sets either consisted of only cells for which lineage tracing data was available (left), all cells (middle), or cells without lineage tracing data (right). Performance metrics evaluated include **f** Pearson r, **g** AUROC, and **h** fraction of test set in which at least 1 simulated cell at the final time point is either classified as a neutrophil or a monocyte over $n = 5$ seeds. In **b**, **f**–**g**, boxplots indicate median (middle line), first and third quartiles (box), and the upper whisker extends from the edges to the largest value no further than $1.5 \times IQR$ (interquartile range) from the quartiles and the lower whisker extends from the edge to the smallest value at most $1.5 \times IQR$ of the edge, while data beyond the end of the whiskers are outlying points that are plotted individually as diamonds. In **h**, bar plots show the average fraction of cells with error bars representing the 95% CI.

we modified an approach described by Schiebinger et al.[8] to use KEGG annotations of cell cycle and apoptosis genes which were also highly variable in the dataset to estimate the number of descendants a cell is expected to have. These estimates correlated weakly but significantly ($r = 0.207$, $p \ll 1e-45$) with the empirical rates calculated from the lineage tracing data (Fig. 2c). We compared models fit to the same set of cells using either our gene-expression-derived or empirical proliferation estimates. We found that models incorporating gene-expression-derived estimates achieved $r = 0.399 \pm 0.025$ and $AUROC = 0.725 \pm 0.008$, a slight improvement over empirical proliferation rate based models (Fig. 2f, g).

**Generative models can simulate trajectories for cells not observed during training.** We next hypothesized that PRESCIENT should be able to predict the fate of cells not observed during training. We expect that the model has learned a good approximation of the underlying potential function from the training data and hence should be able to generalize to unseen data points. To test our hypothesis, we used our proliferation estimates to fit models to all cells with and without lineage tracing data.

We found that model performance was similar when the cells in the test set were included in the training dataset ($r = 0.391 \pm 0.035$, $AUROC = 0.723 \pm 0.013$), and when they were not ($r = 0.407 \pm 0.019$, $AUROC = 0.727 \pm 0.014$) (Fig. 2f, g). Furthermore, although performance as measured by correlation and AUROC is similar to when models were fit only on cells with lineage tracing data, the fraction of test set cells for which the model predicted at least one cell entering a neutrophil or monocyte fate increased slightly from $0.74 \pm 0.01$ to $0.80 \pm 0.03$ (all cells) and $0.83 \pm 0.02$ (only cells without lineage tracing data) (Fig. 2h), suggesting that the models did benefit from observing more data.

**PRESCIENT enables in silico simulations of perturbed cell profiles.** The ability to simulate trajectories for unobserved cells allows the model to make predictions of the fate distribution outcome of cells with perturbed gene expression profiles. We demonstrate this ability on two model systems drawn from published studies with time-series scRNA-seq measurements of differentiation. For these experiments, we first simulate trajectories for an unperturbed initialization (Fig. S2d, e). We then introduce perturbations to the same initial cell population by introducing different levels of overexpression or knock-down of genes that have been reported in the literature to modulate cell

fate outcome (Methods). These perturbations can involve multiple genes, or be introduced at different time points, or in different starting populations. The resulting gene expression profile is then transformed into PCA space to initialize simulations of perturbed cell trajectories.

**PRESCIENT predicts expected changes in cell fate when perturbing transcription factors involved in the regulation of hematopoiesis.** We hypothesized that a PRESCIENT model trained on the Weinreb et. al. dataset should be able to recapitulate the effects on cell fate when perturbing transcription factors (TFs) known to be involved in regulation of neutrophil or monocyte differentiation. (Figs. 3a, b, S3a, b). In particular, we focus on a set of TFs previously identified to be potentially antagonistically correlated with either monocyte and neutrophil fate in progenitor cells by MetaCell analyses and CRISPR-seq experiments of a haematopoietic stem cell dataset[17], many of which are supported by existing literature[18–21].

We first introduced perturbations to *Lmo4*, *Cebpe*, *Mxd1*, and *Dach1*, TFs previously identified to be involved in granulopoiesis, the production of mature neutrophils. As expected, we observed that down-regulation of these TFs led to a relative decrease in the fraction of neutrophils while up-regulation of these TFs led to a relative increase in the fraction of neutrophils with respect to the unperturbed population. We next perturbed TFs involved in monocyte development, including *Irf8*, *Irf5*, *Klf4*, and *Nr4a1*, and observed similar results (Fig. 3c).

We next tested if different magnitudes of the perturbation had different effects (Methods). We found that increasing the magnitude of the perturbation resulted in larger changes in the relative cell fractions (Fig. 3d, e). Furthermore, while individual perturbations resulted in a mixture of significant and non-significant changes in the final neutrophil populations, ensemble perturbations consistently resulted in significant changes ($p < 0.05$; Fig. 3f, g). We also tested multiple sets of randomly selected non-TFs to ensure the changes in final cell fate were not simply a result of perturbations causing random model changes. These randomly selected non-TFs do not result in an observed shift in neutrophil and monocyte fates in the final time point (Fig. 3h), suggesting that our model is robust to random effects.

**PRESCIENT predicts expected outcomes of transcription factor perturbations in endocrine induction introduced at different timepoints and developmental stages.** We next applied PRESCIENT to a 7 time-point scRNA-seq time course of another

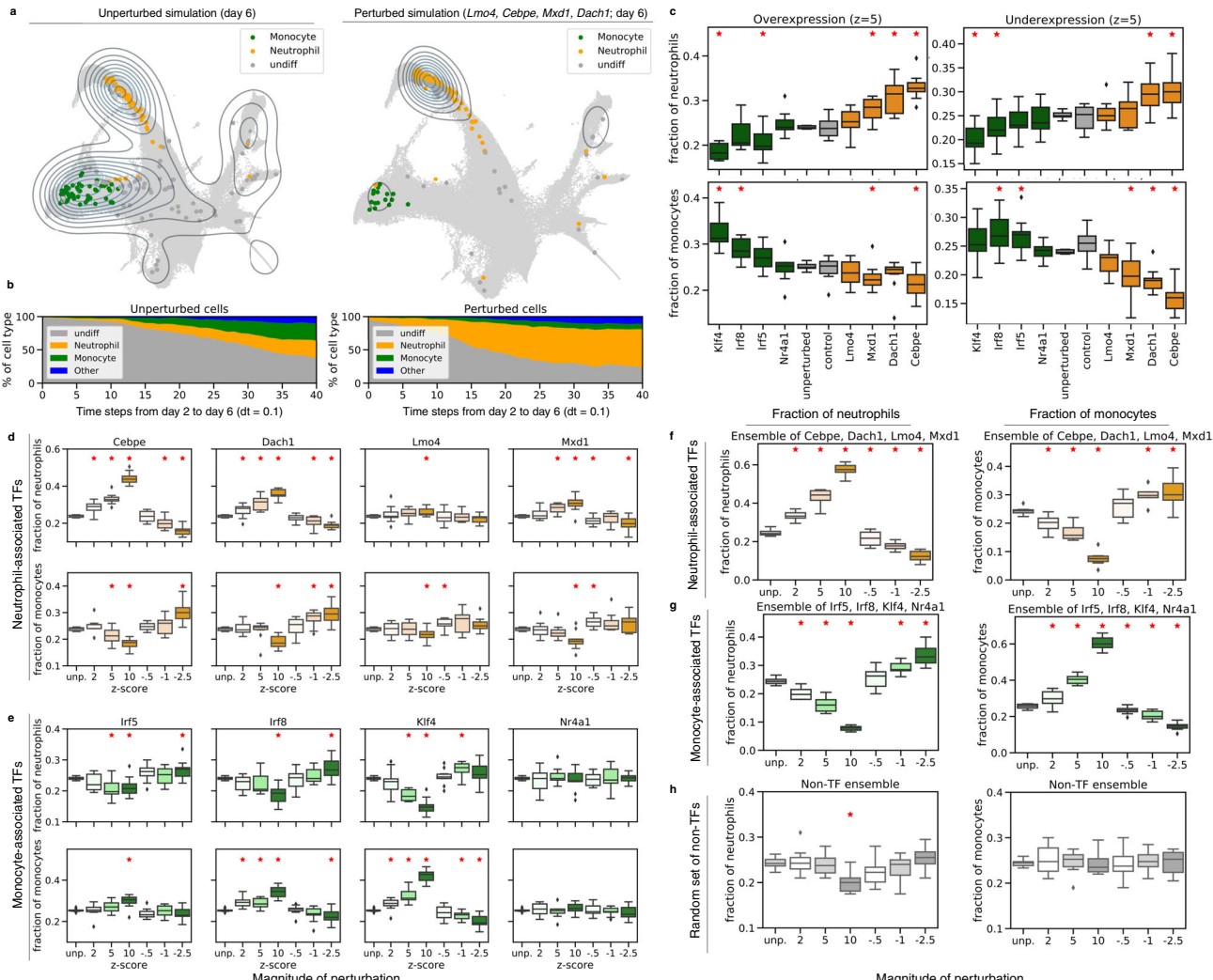

**Fig. 3 In silico perturbations of hematopoiesis results in expected shifts in fate distribution. a** The distribution of cells at the final time-point generated by the model initialized with unperturbed cells (left) and cells with perturbations of *Lmo4, Cebpe, Mxd1*, and *Dach1* upregulated ($z = 10$) during neutrophil differentiation (right). **b** Proportions of generated cell types from day 2 to 6 initialized with unperturbed cells (left) and cells with perturbations of transcription factors upregulated during neutrophil differentiation (right). **c** Fraction of neutrophil and monocyte cells at final time point with single-gene perturbations. Transcription factors involved in monocyte development are indicated in green, while transcription factors involved in neutrophil development are indicated in orange. Control genes (in gray) indicate experiments when perturbing genes from a random set of non-TFs as in (**h**) **d**, **e**, Individual genetic perturbations made to transcription factors involved in neutrophil and monocyte differentiation have an increased effect at higher dosages. **f–g** Ensemble perturbations of transcription factors involved in neutrophil (orange) and monocyte (green) differentiation have a stronger effect. **h** Ensemble random perturbations of non-transcription factors without proliferative signatures (gray). In **c–h**, boxplots are of randomly initialized unperturbed vs. perturbed simulations ($n = 10$) with 200 cells for each initialization, and red asterisks indicate Welch's independent two-sided *t*-test at $p < 0.05$. Boxplots indicate median (middle line), first and third quartiles (box), and the upper whisker extends from the edges to the largest value no further than 1.5 × IQR (interquartile range) from the quartiles and the lower whisker extends from the edge to the smallest value at most 1.5 × IQR of the edge, while data beyond the end of the whiskers are outlying points that are plotted individually as diamonds.

well-characterized differentiation system, the production of pancreatic islet cell types in vitro[22]. This dataset did not include lineage tracing measurements.

We first hypothesized that PRESCIENT should be able to recapitulate the effects on cell fate when perturbing TFs known to be involved in the regulation of endocrine induction and specification in the starting population (Fig. 4a, b). Previous work has shown that *NEUROG3* and *NKX6* activation is associated with the endocrine lineage, while *PTF1A* and *HES1* is associated with the exocrine lineage[23–28]. When introducing ensembled in silico perturbations of *NEUROG3* and *NKX6.1* at day 0, we observe an increase in endocrine cell types that scales with the magnitude of perturbations ($p < 0.05$) with a reciprocal

decrease in exocrine cell types. *PTF1A* and *HES1* overexpression results in the opposite effect (Fig. 4c). We also observed corresponding decreases in endocrine and exocrine cell proportions with knock-downs of *NEUROG3/NKX6.1* and *PTF1A/HES1*, respectively (Fig. S4a, b).

We next simulated the effects of TFs involved in the specification of cell types post-endocrine induction, where *ARX* and *PAX4* have previously been shown to have an antagonistic effect on the specification of endocrine precursors to α and β-cell fate, respectively[24,29–31]. Overexpression of *ARX* in progenitor cells at day 0 resulted in significant increases in the final proportion of α cells at the expense of β cells (Fig. S4c). The same overexpression of *PAX4* resulted in a significant increase in the

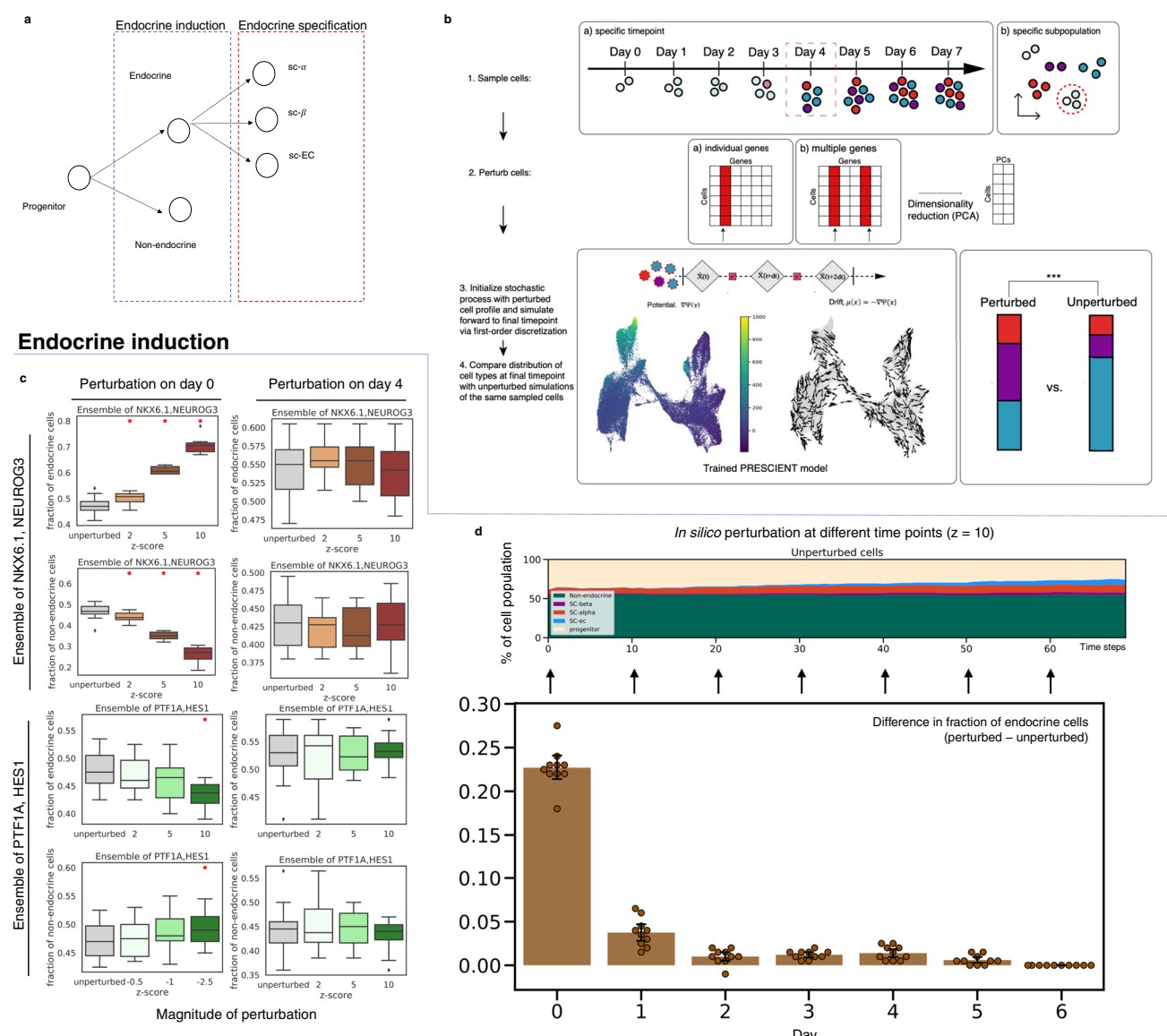

**Fig. 4 PRESCIENT predicts expected temporal dynamics of in silico perturbations of the endocrine/exocrine axis. a** Simplified model of in vitro endocrine induction. Progenitor cells first bifurcate along the endocrine-exocrine axis, and then are specified into more specific pancreatic cell fates. **b** Schematic of steps for different perturbational experiments. First, cells are sampled from either a specific timepoint in the scRNA-seq timecourse or by a specific subtype label. Next, either individual genes or ensembles of genes are perturbed by setting target genes to a higher or lower z-score in scaled gene expression space. This profile is then reduced to principal components. These perturbed cells are then used to initialize a simulation of an already trained PRESCIENT stochastic process, which is simulated forward to the final timepoint. Finally, a pre-trained ANN classifier is used to evaluate the cell-type distribution at the final time-step of the simulation. This distribution is then compared to an unperturbed simulation via a two-sided paired t-test. **c** Final fractions of endocrine and exocrine cells as a result of ensembled perturbations of endocrine- and exocrine- associated TFs (top, bottom) on day 0 and day 4 (left, right) with increasing magnitude. Boxplots (as described previously) are of randomly initialized unperturbed vs. perturbed simulations ($n = 10$) with 200 cells for each initialization, and red asterisks indicate Welch's independent two-sided t-test at $p < 0.05$. **d** In silico perturbations are introduced at different time points to the corresponding unperturbed population. The different outcomes of the cell type of interest are then calculated as the difference in fraction at the final time point starting from the perturbed and unperturbed populations. Bar plots show average differences over $n = 10$ randomly sampled cell populations of 200 cells each from each timepoint with error bars representing the 95% CI.

final proportion of β cells ($p < 0.05$) at the expense of α cells (Fig. S4c).

To demonstrate the scale at which in silico experiments can be conducted, we next did a larger screen of 200+ TFs (Fig. 5a–c). At FDR < 0.01 and $\log_2(FC) > 0.5$, we found that this unbiased screen identified 10 (α-specification increase) and 18 (β-specification increase) TFs that significantly increased final cell fate distribution (Fig. 5a). This includes several known fate-specific factors, such as *ARX* and *IRX2* for α-cells[30,32] and

*NKX2.2*, *NKX6.1*, *PAX4*, *PDX1*, and *MAFA* for β-cells[33–37]. We also identified factors common to both cell lineages, including *MAFB*, *PAX6*, and *ISL1*. *PAX6* has previously been reported to be associated with both α and β-cell fate, but Pax6 mutant mice show a more significant decrease in α-cells, which we also observed[38,39]. In addition, we identified 19 TFs for specification of the less well-studied enterochromaffin (SC-EC) (FDR < 0.01 and $\log_2(FC) > 0.5$) identified by Veres et al., hence showing how PRESCIENT can be used to generate hypotheses for genetic

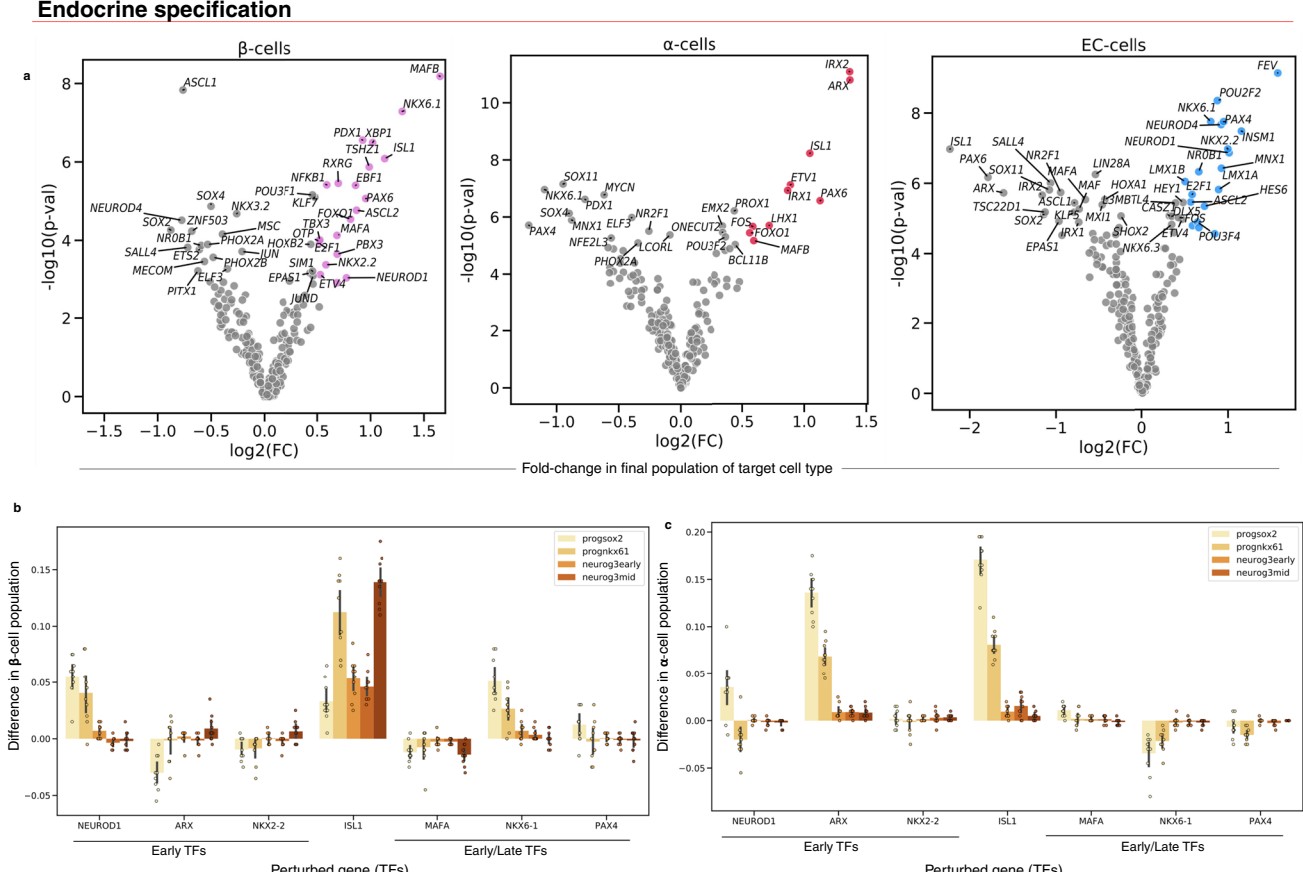

**Fig. 5 PRESCIENT predicts the outcome of transcription factor perturbations in a large perturbational screen as well as the different effects of perturbations in early progenitors vs. cells further along endocrine induction. a** Perturbational screen ($z = 5$) of all the TFs in the highly variable gene set of in vitro β-cell differentiation. The x-axis is the $\log_2$ fold-change (FC) of the final cell-type fraction of the target cell fraction between perturbed and unperturbed simulations. The y-axis is the $-\log_{10} p$ values of two-sided paired $t$-tests between target cell fate outcomes between unperturbed and perturbed simulations over $n = 10$ randomly sampled starting populations at the final time step consisting of 200 cells each. Points are colored if they are a hit (FDR < 0.01 and log2(FC) > 0.5) and β-cell fractions are shown in purple, α-cells in red, and EC-cells in blue. **b** Difference in final β-cell populations when introducing perturbations ($z = 5$) in different cell populations for different TFs. **c** Difference in final α-cell perturbations when introducing perturbations ($z = 5$) in different cell populations for different TFs. In **b**, **c**, different starting populations correspond to cell stages as labeled by Veres et al.: SOX2+ progenitors (progsox2), NKX61+ progenitors (prognkx61), and cells with early/middle/late NEUROG3 signatures (neurog3early, neurog3mid, neurog3late, respectively). Bar plots show average paired differences over $n = 10$ randomly sampled cell populations of 200 cells from each early subpopulation with error bars representing the 95% CI. Figure S4e, f show boxplots of this data.

screens (Fig. 5a). Mean expression of TFs on day 0 was not correlated with predicted $\log_2$(FC) of cell fraction, showing that we could predict the effects of TFs even if their relative expression was low on day 0 (Fig. S4d).

We next asked if PRESCIENT could recapitulate the known timing of TFs during endocrine induction by introducing perturbations to the same endocrine/exocrine axis TFs described above at multiple timepoints (Methods). Endocrine/exocrine induction is known to occur early in the time-course, and the results corroborate this as early perturbations to *NKX6.1* and *NEUROG3* ($z = 10$) result in an increase in endocrine cells, but this effect diminishes with perturbations induced at later time points (Fig. 4d).

PRESCIENT also enables perturbations of cells sampled from different starting cell populations along a given differentiation trajectory. To demonstrate this, we introduced perturbations of selected TFs from the above screen to cells sampled from different stages of the endocrine induction pathway as labeled by Veres et al. (Fig. 5b, *NKX2.2, NKX6.1, PAX4,* and *ARX* are found early in endocrine specification and are often the first signal of the production of specific terminal endocrine cell fates[23]. We found

that perturbations of these TFs in pancreatic progenitors result in a significant increase in final β-cell proportion ($p < 0.05$) and this effect is minimal in cells further along the endocrine induction pathway (Figs. 5b, S4e, f). In contrast, *PDX1* has been shown to continue to promote β-cell neogenesis late into the endocrine induction pathway[40]. We show that perturbations to *PDX1* increase the fraction of β-cells in both progenitor cells and cells with late *NEUROG3* expression (neurog3late), recapitulating the multiple contexts in which perturbations to *PDX1* can modulate endocrine fate. Similarly, *MAFA* and *MAFB* are known to continue to modulate fate late into endocrine induction[37,41], and we show that perturbations to both *MAFA* and *MAFB* result in increases in β-cell proportion when introduced to both early pancreatic progenitors and cells in late endocrine induction. While *ISL1* has been previously reported to stimulate islet cell production, the timing of *ISL1* activation is less well characterized. Our model suggests that *ISL1* has a role in β-cell specification in both early pancreatic progenitor phase and late in the endocrine induction pathway.

Finally, we observed that modulation of α-cell fate largely occurs via perturbations induced in the pancreatic progenitor

phase of the stage 5 differentiation protocol, with diminished effects of all endocrine-specification TFs in cells with early/mid/late *NEUROG3* expression (Fig. 5c). This could suggest that α-cell determination occurs early in the differentiation protocol. We again observe similar outcomes with in silico knockdown experiments (Fig. S4b, e) as well as no effect of perturbations to randomly sampled non-TF genes that are not involved in apoptotic/proliferative signatures (Fig. S5).

## Discussion

PRESCIENT is a generative modeling framework for learning potential landscapes from population-level time series scRNA-seq data. PRESCIENT marks a departure from the predominant methods for analyzing scRNA-seq studies of cellular differentiation. Computational methods for lineage inference have been dominated by pseudo-time approaches that do not attempt to model the stochastic or dynamic nature of cell fate determination. More recent fate prediction methods either summarize observations of the emergent process or suffer from modeling limitations (see Introduction)[8,10]. However, we have demonstrated an important predictive advantage to fully generative models that seek to describe the underlying differentiation landscape. After the model has learned the landscape, it can generate trajectories for unseen data points.

We can hence interrogate PRESCIENT to propose hypotheses for possible perturbations. We show that inducing perturbations in well-studied regulatory genes of hematopoiesis and β-cell differentiation result in expected changes in fate outcome. We also show that we can model the differentiation outcomes of complex perturbations consisting of multiple genes, or at different time points or in different starting populations. This enables large, combinatorial in silico experiments that can help limit the number of in vitro experiments needed to achieve a desired cell fate. PRESCIENT can be used to identify targets for genetic- and small molecule-based screens and aid the design and fine-tuning of new directed differentiation and reprogramming protocols[42–44]. We show an example of this type of unbiased, large-scale screen for in vitro β-cell differentiation in which we perturbed 200+ TFs and identified target genes that could cause significant shifts in β-cell, α-cell, and EC-cell fates. While this work was limited to TFs to show the utility of the method, non-TF targets, such as signaling pathway effectors, can also be tested using PRESCIENT. However, the model is subject to constraints and assumptions, for example requiring that the final time point of the dataset be at steady-state. The model also improves when incorporating growth rates, and would likely benefit from better estimates of proliferation. There also remain challenges to confidently suggest gene sets for experimental perturbation. One problem is that information is lost about individual genes when transforming data into PCA space, and lowly-expressed genes important to cell fate decisions may be dropped altogether in the scRNA-seq data. This can be addressed by methods that have proposed ways to directly generate gene expression counts from latent cell states[45], or by approaches that model perturbation profiles in the original gene expression space[13,46]. These profiles can then be used to initialize PRESCIENT simulations or even extend PRESCIENT to include end-to-end encoding. Another problem is that the association of certain genes with specific cell fates does not necessarily imply causality.

Future extensions of PRESCIENT would accommodate other data or modeling approaches. For example, PRESCIENT's objective can be modified to maximize the likelihood of observing individual trajectories given lineage tracing data. Further, first-principle approaches to modeling dynamics such as RNA velocity are complementary approaches that are non-generative but can be used to constrain PRESCIENT models, e.g. RNA velocity has been proposed for constraining flows across timepoints to local velocities within time points[47–49]. We expect that integrating additional sources of information or increasing sampling density of timepoints should improve the quality of the underlying landscape inferred.

## Methods

**Identifying the latent dynamics of cellular differentiation.** Following Hashimoto et al.[12] we model cellular differentiation as a diffusion process X(t) given by the stochastic differential equation

$$dX(t) = \mu(X(t))dt + \sqrt{2\sigma^2}dW(t) \qquad (1)$$

where X(t) represents the k-dimensional state of a cell at time t, $\mu(X(t))$ is a drift term representing the force acting on a cell given its state, and W(t) corresponds to unit Brownian motion. In particular, the drift function is defined to be the negative of the gradient of a potential function, $\mu(\mathbf{x}) = -\nabla\Psi(\mathbf{x})$, such that intuitively, the potential function $\Psi(\mathbf{x})$ can be thought of as inducing a gradient field driving cells from regions of high potential to low potential. Within the conceptual framework of Waddington's epigenetic landscape, this potential function corresponds to the height of the landscape. This process can be simulated via first-order time discretization

$$X(t + \Delta t) = X(t) + \mu(X(t))\Delta t + \sqrt{2\sigma^2\Delta t}Z(t) \qquad (2)$$

where Z(t) are i.i.d. standard Gaussians. This converges to the diffusion process as $\Delta t \to 0$.

We define the marginal distribution at time t to be $\rho(\mathbf{x}, t) = P(X(t) = \mathbf{x})$. The inference task identifies the potential function $\Psi(\mathbf{x})$, and hence the underlying drift function $\mu(\mathbf{x})$, given only samples from the marginal distribution $\{x(t)_i \sim \rho(\mathbf{x}, t)_i \in \{1...m_t\}, t \in \{1..n\}\}$, where $m_t$ is the number of cells sampled at time t and n is the number of time points where data was observed. In practice, this data corresponds to gene expression profiles of cells sampled over the course of a time-series experiment.

Inference proceeds by finding the potential function $\Psi$ in a family of functions K that minimizes the objective

$$\min_{\Psi \in K}\left[\sum_{i=1}^{n} W_2(\hat{\rho}(t_i, \mathbf{x}), \rho_\Psi(t_i, \mathbf{x}))^2\right] + \tau\sum_{j=1}^{m_n}\frac{\Psi(\mathbf{x_j})}{\sigma^2} \qquad (3)$$

where $\sum_{i=1}^{n} W_2(\hat{\rho}(t_i, \mathbf{x}), \rho_\Psi(t_i, \mathbf{x}))^2$ is the Wasserstein distance between the empirically observed distribution $\rho(t_i, \mathbf{x})$ and the predicted distribution for a candidate potential function $\rho_\Psi(t_i, \mathbf{x})$, and $\tau$ is a parameter controlling the strength of the entropic regularizer. To motivate this loss metric, it is helpful to compare to a case in which we can actually observe ground-truth trajectories of a diffusion process X(t), in which case prediction error can be directly measured as the Euclidean distance between observed points along the trajectory and samples from predicted distribution of X(t) under the model. Wasserstein distance is the direct analog of Euclidean distance when instead considering cross-sectional observations of indistinguishable particles along trajectories of a given diffusion process[50]. In our case, each time point in longitudinal scRNA-seq is a cross-section of cell populations along multiple differentiation trajectories.

**Incorporating cell proliferation.** Using notation from Feydy et al.[51] computing the Wasserstein distance involves solving the optimization problem

$$\min_{<\pi,\mathbf{C}>}\sum_{i,j}\pi_{i,j}C_{i,j}\text{ s.t.}\forall i, j, \pi_{i,j} \geq 0, \sum_{i=1}^{N}\pi_{i,j} = \alpha_i, \sum_{j}^{M}\pi_{i,j} = \beta_j \qquad (4)$$

where $\pi_{i,j}$ is an optimal transport plan mapping points from the source measure to the target measure, $C_{i,j} = ||\mathbf{x_i} - \mathbf{y_j}||^2$ is the squared euclidean distance between point $\mathbf{x_i}$ and point $\mathbf{y_j}$, and $\alpha_i$ and $\beta_j$ are positive weights associated with the samples $\mathbf{x_i}$ and $\mathbf{y_j}$. Previous models minimizing the Wasserstein loss as described by Hashimoto et al. as well as Schiebinger et al. set the weights $\alpha_i$ and $\beta_j$ to be constant, hence assuming uniform sampling over the source and target densities. We incorporate cell proliferation into the modeling framework simply by setting $\alpha_i$ to the number of descendants the cell $\mathbf{x_i}$ is expected to have, where in this case $\mathbf{x_i}$ corresponds to samples from the predicted population. The intuition behind this is that a cell that has a larger number of descendants should be mapped to a larger number of cells at the later time point. Weights associated with the empirical population $\beta_j$ remain constant. Note that in contrast, Schiebinger et al. incorporate cell proliferation by instead using the cell proliferation rates to weaken the marginal constraints, hence reframing the problem as unbalanced optimal transport.

Where lineage tracing data is available, we define the number of descendants a cell is expected to have at time t to be the number of cells sharing a particular clonal lineage barcode at t divided by the number of cells sharing that clonal lineage barcode at the current time. A pseudocount of 1 is added to allow for dropout.

In the absence of lineage tracing data, we estimate the number of descendants using a modified approach from Schiebinger et al. Let n be the number of

descendants, $b$ be the birth rate, $d$ be the death rate, and $g$ be growth. Then using a birth-death process, for a given clone,

$$n = \exp\left(\mathrm{dt} * (b - d)\right) \tag{5}$$

$$g = b - d = \frac{\log n}{\mathrm{dt}} \tag{6}$$

To estimate the birth and death scores ($s^b, s^d$), we first calculate the mean of the $z$-scores of genes annotated to birth (KEGG_CELL_CYCLE) and death (KEGG_APOPTOSIS). We use these alternative annotations as many of the genes in the original gene sets proposed by Schiebinger et al. were not present in the set of variable genes in the Weinreb et al. datasets. Then, following Weinreb et al., we smooth these scores over the cells using an iterative procedure:

$$\beta s_i + \frac{1}{N_{K_{nn}}} \sum_{k \in K_{nn}} (1 - \beta) s_k \tag{7}$$

where $s_i$ is the score for a given cell $i$ at the current iteration, $s_k$ are the scores for the $k = 20$ neighbors of the cell $i$ computed on euclidean distance in PCA space, and $\gamma = 0.1$. This smoothing procedure is performed over 5 iterations.

Then, the birth and death scores are fit to logistic functions to obtain the birth and death rates,

$$b = L_0 + \frac{L}{\left(1 + \exp\left(-k_b s_b\right)\right)} \tag{8}$$

$$d = L_0 + \frac{L}{\left(1 + \exp\left(-k_d s_d\right)\right)} \tag{9}$$

To simplify the fitting procedure, we reasoned that we might expect $-s_{\min}$ to be close to 1 to flatten outliers. Hence, we set $k$ to be $0.001 = \exp(-k s_{\min})$. Then we set $L_0$ and $L$ based on the expected minimum and maximum growth rates. For the Weinreb et al. dataset, we set $L_0 = 0.3$ and $L = 1.2$, and found the minimum and maximum number of descendants to be similar to that observed in the clonal lineage tracing data at days 4 and 6. We used the same procedure for estimating growth rates on the Veres et al. dataset.

**Model implementation and optimization**. We use deep learning methods to learn the parameters of the potential function $\Psi(\mathbf{x})$. Automatic differentiation can be used to evaluate the drift function $\mu(\mathbf{x}) = -\Delta\Psi(\mathbf{x})$ without having to derive the analytical form. Then, by retaining the computation graph, losses can be calculated with respect to the drift function while gradients are accumulated with respect to the potential function. This allows for more flexible parameterizations of the drift function. For all models, we set $dt$ and $\sigma$ to be 0.1, and $\tau$ to be 1e−6.

Optimization of $\Psi(\mathbf{x})$ was performed using the Adam optimizer with a batch size of 1/10th the size of the training set. Fully connected 2-layer, 400-unit models using softplus as the activation function was used as the architecture for all models. This architecture was chosen using performance on the held-out recovery task on the Weinreb et al. dataset (Supplementary Note, Figs. S1, S3). Models were pre-trained as previously described by Hashimoto et al. by first optimizing only the entropic regularizer via contrastive divergence using stochastic gradient descent with a learning rate of 1e−9. Then, training proceeds at the learning rate described below, with a scheduler that multiplied the learning rate by 0.9 every 100 iterations, and using gradient clipping with a max norm of 0.1. The Wasserstein error is approximated using a multi-scale Sinkhorn algorithm as implemented by the GeomLoss library (v0.2.3), with a scaling of 0.7 and a blur of 0.1. The GeomLoss library allows for efficient, stable computation of gradients that bypasses naive backpropagation[51].

All models were trained using a single NVIDIA Titan RTX GPU (24GB RAM). All forward simulations, including perturbation simulations, were run on either a single NVIDIA Titan RTX GPU (24GB RAM) or a single GeForce GTX 1080 Ti (11GB RAM). Runtime estimates for training and simulations are provided (Fig. S1f, g). Training runtimes were computed using two different individual GPUs on dataset sizes ranging from 1000 to 130,000 cells and the range of training times are ~13 min–1 h. Training with GPU acceleration is necessary for training times to be tractable, and is a one-time cost. For those without access to GPUs, we recommend utilizing Amazon Web Services or Google Cloud GPU resources for training, which should be very low-cost for training a PRESCIENT model. Simulation runtimes were computed by measuring the time to run forward simulations of random initializations of $n = 400$ for a fixed number of steps with 1, 10, 100, 1000, and 10,000 random initializations serially. Simulations can tractably be generated using CPUs or GPUs.

All nearest-neighbors calculations (for eg. for cell-type classification) were calculated using the approximate nearest-neighbors library annoy (https://github.com/spotify/annoy).

**Visualization of potential landscapes**. The potential landscape learned by the model was visualized by evaluating the potential on a uniform grid in UMAP space.

The drift function is visualized as unit arrows, where the point of origin is given by the same grid and the vector is given by the drift evaluated at the point of origin.

**Preprocessing of existing scRNA-seq datasets**. Preprocessed data for the Weinreb et al. experiments was downloaded from https://github.com/AllonKleinLab/paper-data/blob/master/Lineage_tracing_on_transcriptional_landscapes_links_state_to_fate_during_differentiation/README.md (commit: d8f0969)[3]. The set of highly variable genes was determined as by Weinreb et al. by first filtering for highly variable genes, and then excluding genes correlated with cell cycle (SPRING, commit: a37bbd0). Normalized gene expression for variable genes was scaled and projected to 50 dimensions via PCA, which was then used as input to the modeling framework. For experiments evaluating the model on the held-out time point, preprocessing was fit to only the training set consisting of days 2 and 6, and then used to transform all data including day 4. For all other experiments, preprocessing was fit to all data across time points. All visualizations using umap were fit with 30 neighbors.

Data for the Veres et al. experiments was downloaded from GEO (GSE114412)[22]. Raw counts were first pre-processed using the standard Seurat pipeline (v3.1.5)[52] to obtain normalized counts. For feature selection, genes were first filtered for those observed in at least 10 cells. Then, the 'FindVariableFeatures' function was used to identify the top 2500 most variable genes. Scaled gene expression was then computed as for the Weinreb et al. dataset. For projection into PCA space (30 PCs), the variable gene set was filtered again to remove genes correlated with TOP2A ($r > 0.15$), as described by Veres et al. This was used as input to the modeling framework, and for visualization via UMAP. For estimation of proliferation rates, the full variable gene set was used (Fig. S3a, b, e, f).

**Experiments on recovery of a held-out time point**. For comparison to Waddington-OT (WOT)[8], which uses held-out recovery (interpolation) as a benchmark, we fit models via pre-training on day 6 and then trained on days 2 and 6 for evaluation on day 4. Models were fit using only the subset of data for which lineage tracing data was available to enable comparison of models incorporating empirically derived cell proliferation rates. All models were trained for 2500 epochs.

To evaluate the models at a given time point, 10,000 cells were sampled at day 2 with replacement according to the expected cell proliferation rate. Then, the model was used to sample a single trajectory for each of the sampled cells until the time point under evaluation. The Wasserstein distance was then computed between the simulated cell population and the empirically observed cell population. Models were evaluated at day 4 (held-out, testing) and day 6 (training) every 100 training epochs. The testing distance is reported for the epoch with the lowest training error (Fig. S1c).

To compute the linear interpolation baseline, we used Waddington-OT (WOT), which uses a similar optimal transport formulation but lacks an explicit parametric form[8]. WOT enables recovery of a held-out time point via linear interpolation using transport maps built between sets of cells in early and late time points. To run WOT, we used python code available on GitHub (https://github.com/broadinstitute/wot). The input to WOT is a set of time-point labeled gene expression profiles and growth rates and the output is an optimal transport map. The optimal transport map was built with the full set of cells with lineage barcodes from day 2 ($n = 4638$ cells) to day 6 ($n = 29,679$ cells). The empirical proliferation rates derived from clonal expansion of this set of cells from day 2 to 6 were provided to WOT and three growth iterations were permitted. The parameters for building the optimal transport map were as follows: $\lambda_1 = 1$, $\lambda_2 = 50$, $\varepsilon = 1$, $\tau = 10,000$. With the transport map built between day 2 and day 6, 10,000 cells at day 4 were interpolated using the interpolate_with_ot() function from WOT. This maps a point at the midpoint of each of the pairs in the optimal transport map. The testing distance of these interpolated points from the observed day 4 cells was computed as reported above.

**Predicting clonal fate bias**. To predict clonal fate bias, models were first trained on data from all three time points. Models were fit on three sets of data; (a) the subset of cells for which lineage tracing data is available, (b) the subset of cells for which no lineage tracing data is available, and (c) all cells. Models were trained for 2500 epochs and evaluated every 500 epochs. Then, cells were simulated until the final time point via the first-order discretization as parameterized by the trained model.

We evaluated the clonal fate bias metric described by Weinreb et al. on the test set as defined in their paper. The ANN classifier that we used to classify cells as Neutrophil, Monocyte or other at the final time point was fit with 10 trees, 20 neighbors and using Euclidean distance in PCA space (50 pcs). The model was first fit on a random 80% split of the data. When evaluated on the held-out 20% test split, the model achieved a macro-average f1-score of 0.98. Splits were stratified by cell type. The model was then re-fit to the full dataset. After classification, the clonal fate bias was then computed as the number of neutrophils divided by the total number of monocytes and neutrophils. Since the model did not always predict any cell within the 2000 sampled trajectories to be a monocyte or neutrophil, we also added a pseudocount of 1. In those cases, clonal fate bias would hence be 0.5.

We observed variation in predictions made by models at each epoch (Fig. S2a). Since no validation set is available to formulate a stopping criterion, we chose to ensemble predictions over the last 5 epochs evaluated (i.e., epoch 2100, 2200, 2300, 2400, 2500) by taking the mean across the estimated clonal fate bias across those epochs.

In most cases, performance metrics for WOT, PBA, and FateID were calculated using the predictions already pre-computed and made available by Weinreb et al.[3]

**Introducing and evaluating in silico perturbations**. Perturbation experiments were performed similarly to the clonal fate bias experiments, except using perturbed cells as input to the first-order discretization. Generally, perturbations were introduced in silico by setting the scaled normalized expression of target genes to $z$-score values less than 0 for knockdowns and greater than 0 for overexpression. The resulting perturbed gene expression profile was then transformed via PCA into lower-dimensional space for input to forward simulation of the trained model.

For the in vitro hematopoiesis dataset[3], the trained PRESCIENT model used for predicting the effect of in silico perturbations to this dataset was seed 1 and epoch 2500 of a model trained with a neural network architecture of 2 layers of 400 units. For all experiments, 200 undifferentiated cells (annotations from Weinreb et al.) were randomly sampled from day 2 weighted by KEGG-derived growth rate estimates, resulting in biased sampling for actively proliferative cells. These sampled cells were simulated forward 40 steps with a dt of 0.1 to the final time point (day 6). This process was repeated with random initializations of both unperturbed cells and perturbed cells. Cells at the final time point were then classified using the same ANN classifier used for the clonal fate bias experiments. Relevant TFs for the target cell fate were identified by searching the literature for studies that had experimentally verified sets of TFs involved in early cell fate decisions by progenitor populations and the highly variable feature set was filtered for these TFs. Perturbations were focused on monocytes and neutrophils due to the availability of experimentally correlated or confirmed perturbations for these two cell types and the focus of neutrophil/monocyte fate in the Weinreb et al. lineage tracing data. Neutrophil-associated transcription factors were *Lmo4, Cebpe, Mxd1,* and *Dach1*. Monocyte-associated transcription factors were *Irf8, Irf5, Klf4,* and *Nr4a1*. First, the effect of perturbations to individual genes were tested by perturbing each target gene with a $z$-score of $-2.5$ for knockdown and 5 for overexpression. To test if there was a significant shift in neutrophil/monocyte cell fractions at the final time point, Welch's independent two-sided $t$-tests were performed between unperturbed simulations and perturbed simulations for each target TF. Next, the effect of perturbational magnitude was evaluated by introducing perturbations of $-2.5, -1, -0.5, 2, 5,$ and 10 to each target gene individually. The same statistical test was performed in comparison to unperturbed simulations. Next, we tested the effect of ensembled perturbations to cell fate outcome by perturbing sets of TFs with $z$-scores of $-2.5, -1, -0.5, 2, 5,$ and 10. As a control, the ensembled perturbation was repeated with following randomly selected non-TF control genes: *Gch1, Pfn2, Dhrs2, Traf1, Lrrk2, Lgmn, Il13,* and *Sgk1*. The same statistical test was performed in comparison to unperturbed simulations.

For the in vitro beta-cell differentiation dataset[22], we trained PRESCIENT models with a learning rate of 0.001 to prevent model divergence. All other parameters were as in the models trained for clonal fate bias prediction in the Weinreb et al. dataset. The trained PRESCIENT model used for predicting the effect of in silico perturbations to this dataset was seed 1 and epoch 1500 of a model trained with 2 layers of 400 units. The 400-unit model was chosen as it achieved a lower training distance, and epoch 1500 was chosen based on the training curves to prevent overfitting, since we observed that training performance had appeared to have plateaued by that epoch (Fig. S3c, d). As with the Weinreb et al. dataset, an ANN classifier with 10 trees, 20 neighbors, and using Euclidean distance in PCA space (30 PCs) was trained to predict cell type on a training set of 80%. On a randomly held-out test set of 20%, the model achieved a macro-average f1 score of 0.939 when discriminating between sc-α, sc-β, sc-EC and other cells. The ANN classifier was then re-fit on the full dataset. For time point sampling experiments, 200 cells were sampled from each time point weighted by KEGG-derived growth rates, based on metadata from Veres et al. Cells were iteratively sampled from days 0–6 and simulated forward with a $dt$ (step size parameter of 0.1) to the final time point (day 7) under both unperturbed and perturbed conditions. Perturbations ($z = -2.5, -1, -0.5, 2, 5, 10$) were introduced to sampled cells from each timepoint. To test if there was a significant shift in neutrophil/monocyte cell fractions at the final time point, Welch's independent two-sided $t$-tests were performed between unperturbed simulations and perturbed simulations for each target TF. For cell-type subpopulation experiments, 200 cells were randomly sampled from the SOX2+ progenitor, NKX6.1+ progenitor, and NEUROG3 early/mid/late, weighted by KEGG-derived growth rates. For the screen of 200+ TFs, 200 cells were first simulated without perturbations and the same cells were simulated with perturbations ($z = 5$) of each TF. This was repeated with 10 random initializations. For each TF, to test for a significant change in final cell-type fractions, two-sided paired $t$-tests were conducted between the unperturbed and perturbed simulations. To show that TF expression at static timepoints is not necessarily indicative of fate bias, a control analysis was completed on the Veres et al. 2019 dataset by computing the log-fold change of each TF between cell types

of interest at day 7 and plotted against the log-fold change in cell fraction predicted by PRESCIENT perturbational analysis (Fig. S4d).

**Reporting summary**. Further information on research design is available in the Nature Research Reporting Summary linked to this article.

## Data availability

Data for the Weinreb et al. experiments was downloaded from https://github.com/AllonKleinLab/paper-data/blob/master/Lineage_tracing_on_transcriptional_landscapes_links_state_to_fate_during_differentiation/README.md (commit: d8f0969)[3]. Data for the Veres et al. experiments was downloaded from GEO (GSE114412)[22]. Trained models from this study are available at https://zenodo.org/record/4687963#.YHeXOBNKiU9 or https://www.github.com/gifford-lab/prescient-analysis.

## Code availability

An open-source implementation[53], documentation, and tutorial vignettes of PRESCIENT is available at https://cgs.csail.mit.edu/prescient/. Source code can be found at https://github.com/gifford-lab/prescient. Notebooks to reproduce figures and analyses are available at https://github.com/gifford-lab/prescient-analysis.

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

## Acknowledgements

We thank Jennifer Hammelman and Ernest Fraenkel for helpful discussion and comments. We gratefully acknowledge funding from NIH grants 5 R01 NS109217, 5 R01 HG008754, and 5 R01 HG008363 (D.K.G.) and the National Science Scholarship (PhD) from the Agency for Science, Technology and Research Graduate Academy (G.Y.).

## Author contributions

Conceptualization, methodology, software, validation, formal analysis, investigation, data curation, writing, visualization: G.Y., S.S.; conceptualization, writing, formal analysis, supervision, funding acquisition: D.K.G.

## Competing interests

D.K.G. is a founder of Think Therapeutics. All other authors declare no competing interests.
