## [Peer Review File · Nature Communications]

Reviewers' Comments:

Reviewer #1:

Remarks to the Author:

In this paper, the authors introduced PRESCIENT, a generative modeling framework to learn the differentiation landscape of a population of cells from their associated time-series scRNA-seq data. In terms of the broader context, PRESCIENT joins the ranks of a plethora of computational tools that infer differentiation trajectories during development, and attempt to predict the fates of individual cells. scRNAseq technologies are limited in that they can only capture discrete snapshots of development --- and the field often relies on computational methods to connect them into "continuous movies".

The general idea behind PRESCIENT is that certain types of diffusion processes can be represented as the gradient of a potential. In the context of single cell RNAseq data, gene expression values are used to infer the gradient by modeling the trajectory of a cell as a random walk on the differentiation landscape (known as a diffusion eigenmap). This model allows for the interpretation that cells commit to different fates depending on which path they follow. From a computational point of view PRESCIENT attempts to fit a neural network, specifically a fully-connected 2-layer 400 unit model, to learn the potential gradient function. The resulting model can then be interrogated for follow-up inference and predictive purposes.

I have the following major concerns about this manuscript:

1 I am concerned about the lack of novelty in the proposed approach. Pseudotime analyses are extremely popular, and tools like Waddington-OT and FateID have also attempted to tackle the specific challenge of cell fate prediction, albeit with different approaches, based on optimal transport and random forests respectively. For the prediction of in silico single-cell perturbation responses, there exist some methods like scGEN, which adopt a variational autoencoder approach to this problem. PRESCIENT might be one of the only few methods that can claim to be a one-for-all solution for these tasks and more, but against this background of software packages, has PRESCIENT done enough to stand out?

2 Having said that, one could argue that because the overall field of modeling cell trajectories, predicting cell fates and in silico perturbation is a relatively nascent field, especially in single cell genomics. There might indeed be some value in publishing an alternative tool for users, and with that mind, it should be easy and straightforward for prospective users to download, install and use the tool. Unfortunately, PRESCIENT falls severely short in that regard. The manuscript does not provide a link to any relevant code repository, though a web search has come up with this GitHub page (<https://github.com/gifford-lab/prescient>). Assuming this is the correct code base, we had a difficult time understanding how to run and reproduce the analysis in the paper, which is unacceptable by today's standards. For example, one group of individuals who will likely be keen to try this tool would be experimental biologists looking for potential mutagenesis targets, but sadly, we cannot imagine a scenario where they will be able to get this to work. It is also not clear how scalable or compute intensive PRESCIENT is, though the methods section and GitHub repo explicitly mentioned the usage of NVIDIA top-of-the-line GPUs, suggesting this might not be accessible to a majority of the biology community. Substantial improvements in the user's quality of life are warranted to provide better usability and interpretability.

Specifically, we would like the authors to imagine themselves as day-to-day biologists who wish to do an unbiased in silico perturbation experiment with their single cell data, how would they get started with PRESCIENT? As some suggestions, a fleshed out guide or tutorial is definitely needed, as well as an easy way to download and install all the required dependencies. The availability of computing resources for the general user might also need to be considered, otherwise it really limits how useful the tool really is. All in all, a computational tool is only useful if it is usable, and PRESCIENT certainly

has potential if these concerns are addressed.

3 Due to the number of layers in the neural network, we also anticipate problems in the interpretability of the trained deep learning model. The approach the authors have adopted was to treat PRESCIENT as a black box (as mentioned in line 86-87) to approximate cell fate. It is not immediately clear to us how an average user would initialize an in silico perturbation experiment and “measure how much the perturbation of a single gene would impact the cell fate decision”. A feature of these neural networks is that the bottleneck layers tend to summarise key features of the data, and we feel like the authors should consider using those to visualise how the gene perturbation affects the landscape, rather than just treating it as a predictive model alone.

4 We also have several concerns about the general presentation of text in the manuscript. There are some errors and inconsistencies in the equations, in particular with regards to the use of the del operator. For example, in Fig 2d, both y-axes are labeled with $\nabla \Psi$, although one is supposed to be the potential and the other the drift. This does not seem correct. Similarly, on lines 504 and 575, the drift is written as $\Delta \Psi$ instead of as $\nabla \Psi$. We also find some of the Figure 3-5 too small and illegible, and question what the value is of including all these miniature box plots. Suggestions would be for these errors to be fixed, and to summarize these plots more succinctly in the main text.

Reviewer #2:

Remarks to the Author:

Yeo, Saksena and Gifford present a framework for modeling differentiation trajectories from single cell RNA-Seq data. Although there has been a lot of previous work in this area, this manuscript sets itself apart by modeling the process over real time rather than pseudotime and by being a generative model, thus allowing for interesting in silico perturbation experiments. I believe the main contribution of this manuscript is to extend the previous work of the authors (Hashimoto, Gifford and Jaakkola 2016), incorporating a model for cell proliferation as well as extensively demonstrating and benchmarking the framework of Hashimoto et al. and its applications to the single cell community.

Overall this is an important advance and a very welcome addition to the scRNA-seq analysis toolset with potential for wide use. However, the manuscript text needs some revision as it is currently impenetrable to a wider audience in my opinion. Furthermore, to ascertain the practical relevance of the method, the tool and code require documentation to make them usable by the wider community (<https://github.com/gifford-lab/prescient>), and some method and benchmark details need clarification.

Main Comments:

(1) I think the main practical attractive feature of this framework is the ability to simulate perturbations in time-course experiments, so I will focus here this main comment in this context:

- I wonder why the authors did not choose to perform a transcriptome-wide unbiased screen (they do a 200 TF screen). Is there a theoretical or practical limitation?
- How much time and compute resources does the method require? Can the authors provide a computational benchmark, given certain GPU hardware how does the method perform vs cell number, and perturbation experiment number.
- There are recent attempts addressing similar problems (Lotfollahi et al. Nat Methods 2020), this is mentioned only in passing but further delineation of the differences would be very welcome. Also WOT, which the authors compare to in Figure 2 can provide some predictions of which TFs are important for cell fate - it's not exactly perturbation and so not directly comparable but still appropriate to include WOT results also in one of the TF

case studies such as the one in Figure 5

(<https://nbviewer.jupyter.org/github/broadinstitute/wot/blob/master/notebooks/Notebook-8-predictive-TFs.ipynb>).

- The authors perturb the genes in the space of z-scores but how does the expression level of the genes affect the ability to predict their perturbation effects? Furthermore, the entire model is run in PCA cell space so changes due to specific genes are difficult to disentangle. For example, only the top 2500 variable genes are used to derive the principal components in one part of the study.
- There is a focus on TFs, I guess the motivation is that TFs define the waddington differentiation landscape in principle, but from a performance point of view: what happens if essential genes are perturbed? what happens if signaling molecules are perturbed? Is it appropriate to use this model to study these perturbations even though paracrine effects are not accounted for by the model?
- In figure 4c, what is the observed expression of the TFs at the evaluated time-points? Also in Figures 5a,b.

(2) The main code repository of this work is provided here: <https://github.com/gifford-lab/prescient>, however without any documentation or code comments. This is not about reproducibility of this paper's results (although that's also important) but about whether others can easily test and use the methods for their own data. I strongly encourage the authors to invest the time in providing accessible tutorials and documentation that clearly outline input file formats, expected outputs, what the different functions do and so on. The authors can take the tutorials provided by popular scRNA-seq tools such as Seurat and Scanpy as examples. It's worth the effort because it will increase the impact of this work significantly and would've helped clarify many of the questions below.

(3) In many places in the text, previous specialized knowledge related to specific papers is assumed. This can be improved to make the paper accessible. I would recommend the authors go through and make sure all important concepts are explained in the main text when possible and further elaborated in supplementary notes. I quote a few examples that the authors can choose to revise:

- lines 45,46: please elaborate what the other methods do - what does "modeling capacity" mean? This is important to elaborate in the main text because it's related to how this work is placed within the context of other similar methods. Supp. Note 1 might be good to include in the main text if possible.
- line 49: What is the model in Hashimoto et al. and how exactly does this paper extend it? (one starts to infer this only if we read the supp. methods or Hashimoto et al. The latter nicely explains the motivation but this needs to be elaborated also here.)
- line 53: "cell fate bias". this phrase is repeated often but not clear to me what it means at this stage in the text and whether its meaning slightly changes in different places.
- Section starting at line 118: it wasn't elaborated how cell proliferation is incorporated. perhaps a brief explanation and reference to the supplemental methods is appropriate. Please also see comment (8)

Other comments & questions:

(4) Using correlation as a benchmark is not very convincing (example: figure 2b) - the impressive p-value is likely because there is a lot of data points, not sure this is really indicative of a correlation. Also please double check Figure 2f, and please provide the scatter plots in the supplement.

(5) Wasserstein distance is used during model fitting as the main constraint (minimize the Wasserstein distance between observed data and the model fit, line 521) - but also sometimes as a benchmark (example: Figure 2a). Use as a benchmark requires explaining what it is in the main text and justification why it's a good way to measure performance relative to the model of WOT.

(6) The data is always represented as a UMAP, and the potential function and drift is represented on top of that. Can the authors also evaluate the cell pseudotime using any of the other methods on some of the data and present this, and also plot the cell physical time? this is helpful as it may give an intuition to potential performance difference in practice between the typical approaches and what the authors propose. Also I'm trying to understand if there is a more direct way to represent the potential function, what does the drift direction (arrows) mean in the independently-derived UMAP space? Would be very helpful if this is clarified explicitly in the manuscript.

(7) I wonder how the model performance might be affected by the time sampling density. I imagine at some point, predictions would break as sampling density decreases depending on the complexity of the differentiation process in question. If the authors have evaluated this, it might be good to mention in the discussion since it will help with experiment design considerations if one is planning to use this method.

(8) I have to admit after reading the section starting at line 526 I still did not understand (ie. do not have a clear intuition) how exactly the proliferation rates would affect the objective function on line 521. As this is probably the main extension relative to Hashimoto et al and key to performance improvement (Figure 2), I would like to kindly ask the authors to elaborate this further. Also related to this, it was a bit surprising to me that basing proliferation on ground truth rates did worse than when based on KEGG gene sets (Figure 2f-g).

(9) does the model allow for stationary cells (such as senescent cells that were assayed at $t=0$) or would those correspond to infinite potential? - thank you also for the clarification in the discussion if appropriate.

(10) in practice, sampling a time-course for single cell studies can result in batch effects. were these taken into account during benchmarking? do the authors recommend a certain method for deriving principal components in those cases as input to their model, such as MNN or Harmony?

(11) Figure 4d is mentioned after Figure 5

REVIEWER COMMENTS

Reviewer #1 (Expertise: Analysis of time-resolved scRNASeq data):

In this paper, the authors introduced PRESCIENT, a generative modeling framework to learn the differentiation landscape of a population of cells from their associated time-series scRNA-seq data. In terms of the broader context, PRESCIENT joins the ranks of a plethora of computational tools that infer differentiation trajectories during development, and attempt to predict the fates of individual cells. scRNAseq technologies are limited in that they can only capture discrete snapshots of development --- and the field often relies on computational methods to connect them into “continuous movies”.

The general idea behind PRESCIENT is that certain types of diffusion processes can be represented as the gradient of a potential. In the context of single cell RNAseq data, gene expression values are used to infer the gradient by modeling the trajectory of a cell as a random walk on the differentiation landscape (known as a diffusion eigenmap). This model allows for the interpretation that cells commit to different fates depending on which path they follow. From a computational point of view PRESCIENT attempts to fit a neural network, specifically a fully-connected 2-layer 400 unit model, to learn the potential gradient function. The resulting model can then be interrogated for follow-up inference and predictive purposes.

I have the following major concerns about this manuscript:

1. I am concerned about the lack of novelty in the proposed approach. Pseudotime analyses are extremely popular, and tools like Waddington-OT and FateID have also attempted to tackle the specific challenge of cell fate prediction, albeit with different approaches, based on optimal transport and random forests respectively. For the prediction of in silico single-cell perturbation responses, there exist some methods like scGEN, which adopt a variational autoencoder approach to this problem. PRESCIENT might be one of the only few methods that can claim to be a one-for-all solution for these tasks and more, but against this background of software packages, has PRESCIENT done enough to stand out?

Response:

We have now included a discussion of existing pseudo-time methods in the introduction and how PRESCIENT is different in that it operates in real-time (rather than pseudotime) and is a queryable, parametric model. Pseudotime methods only order cells, don't predict distributions over cell fates, don't predict perturbations and are difficult to compare with lineage tracing methods because they are not generative, i.e. they don't produce distributions of cells over time. “Model-free” or “coupling” models (like WOT and FateID) are limited in utility across datasets and for generalizing differentiation trajectories. For example, these models rely on observed training

points and cannot accommodate out-of-sample gene expression profiles, limiting the ability to predict the trajectories of perturbed cells.

scGen shows the effect of a perturbation on an entire dataset in gene expression space. In contrast, PRESCIENT shows the effect of a perturbation of a specific cell, at a specific time point over an entire differentiation process. scGen does not model real time or distributions of the perturbational effects i.e. given a single cell and the perturbation applied to it, it doesn't give a distribution over the perturbed cell, simply a deterministic shift in the latent space corresponding to a shift in the original space. It is not clear that scGen could be simply applied for this use, and their original paper does not claim this function. scGen could benefit PRESCIENT models, as perturbed gene expression profiles could be more appropriate for initialization of PRESCIENT in perturbation mode but is outside the scope of the study.

2 Having said that, one could argue that because the overall field of modeling cell trajectories, predicting cell fates and in silico perturbation is a relatively nascent field, especially in single cell genomics. There might indeed be some value in publishing an alternative tool for users, and with that mind, it should be easy and straightforward for prospective users to download, install and use the tool. Unfortunately, PRESCIENT falls severely short in that regard. The manuscript does not provide a link to any relevant code repository, though a web search has come up with this GitHub page (<https://github.com/gifford-lab/prescient>). Assuming this is the correct code base, we had a difficult time understanding how to run and reproduce the analysis in the paper, which is unacceptable by today's standards. For example, one group of individuals who will likely be keen to try this tool would be experimental biologists looking for potential mutagenesis targets, but sadly, we cannot

imagine a scenario where they will be able to get this to work. It is also not clear how scalable or compute intensive PRESCIENT is, though the methods section and GitHub repo explicitly mentioned the usage of NVIDIA top-of-the-line GPUs, suggesting this might not be accessible to a majority of the biology community. Substantial improvements in the user's quality of life are warranted to provide better usability and interpretability.

Specifically, we would like the authors to imagine themselves as day-to-day biologists who wish to do an unbiased in silico perturbation experiment with their single cell data, how would they get started with PRESCIENT? As some suggestions, a fleshed out guide or tutorial is definitely needed, as well as an easy way to download and install all the required dependencies. The availability of computing resources for the general user might also need to be considered, otherwise it really limits how useful the tool really is. All in all, a computational tool is only useful if it is usable, and PRESCIENT certainly has potential if these concerns are addressed.

Response:

We agree that the original codebase was difficult to use. We have made the following adjustments:

- We packaged PRESCIENT as a PyPI package that can be easily installed and run.
- We now have a website <https://cgs.csail.mit.edu/prescient/> with complete command line documentation, quickstart, analysis vignettes, and input formats.

- We refactored the codebase to make the scripts dataset agnostic and specified a standardized input format for training models and running perturbation simulations with already trained models.
- We have made a separate repository (<https://github.com/gifford-lab/prescient-analysis>) containing notebooks to reproduce the figures presented throughout the paper. In addition, all training data, trained models, perturbational outcomes in the paper have been uploaded to a Dropbox (<https://www.dropbox.com/home/prescient-data>).
- We agree that access to GPUs may be a problem regarding use by biologists, and we now include a runtime analysis in the supplement that shows training PRESCIENT on a GPU is tractable (average training time on full datasets ~30min-1hr) (Fig. S1f-g). To address concerns about computational feasibility, we have now included a recommendation for those without GPU access to utilise Amazon Web Services or Google Cloud GPU services to train a PRESCIENT model. This should not be prohibitively expensive, as runtimes on GPUs are on the order of minutes and hours, not days. We also want to emphasize that training is a one-time cost and perturbational simulations are fast and efficient, even on a CPU (Fig. S1f-g).

We have also provided this information in the code availability section.

3. Due to the number of layers in the neural network, we also anticipate problems in the interpretability of the trained deep learning model. The approach the authors have adopted was to treat PRESCIENT as a black box (as mentioned in line 86-87) to approximate cell fate. It is not immediately clear to us how an average user would initialize an in silico perturbation experiment and “measure how much the perturbation of a single gene would impact the cell fate decision”. A feature of these neural networks is that the bottleneck layers tend to summarise key features of the data, and we feel like the authors should consider using those to visualise how the gene perturbation affects the landscape, rather than just treating it as a predictive model alone.

Response:

We agree that interpretability of deep learning models is a concern when developing methods for biological discovery. But we argue that the framework in which we fit a neural network - as a parameter of a queryable stochastic ODE - is itself a move towards a much more interpretable cellular trajectory analysis (and is, in fact, a key motivation of our work). In this work, we try to shift from the statistical ordering of cells (i.e pseudotime) or the task of simply predicting cell state from gene expression profiles (e.g many existing machine learning approaches) to a parametric and generative model that simulates a distribution at each time step that can be visualized and allows for simulations with out-of-sample cells. This is akin to traditional systems modeling in engineering (systems of ODEs, PDEs, etc), which allow for simulations of perturbations as a form of interpretability.

To address your specific points:

The neural network that gives the drift parameter of the stochastic ODE diffusion process is actually quite shallow (1 or 2 layers) relative to modern high capacity machine learning approaches. That being said, we agree that some interpretability methods are difficult to apply to

our framework because of its structure as a diffusion process. For example, gradient-based interpretability methods like saliency mapping and integrated gradients are not directly applicable. However, another common interpretability method is *in silico* saturation mutagenesis (ISM), which is a feature attribution technique for inferring contributions of all inputs to the model's predicted output via perturbing each input (Zhou and Troyanskaya 2015; Kelley et al. 2016; Nair et al. 2020). This is effectively what we do for our machine learning framework in the context of scRNA-seq initial cells.

We do agree that there are some interpretability limitations with respect to our ability to operate on the original gene expression input space. The transformation from gene to PCA space and inverse transformation is too lossy to retain the correct resolution of information to draw conclusions. The difficulty of operating on gene expression space itself is also observed by other comparable models like WOT and scGen that both operate on PCA space and the latent space of an autoencoder derived from the top N highly variable genes, respectively.

4. We also have several concerns about the general presentation of text in the manuscript. There are some errors and inconsistencies in the equations, in particular with regards to the use of the del operator. For example, in Fig 2d, both y-axes are labeled with $\nabla \Psi$, although one is supposed to be the potential and the other the drift. This does not seem correct. Similarly, on lines 504 and 575, the drift is written as $\Delta \Psi$ instead of as $\nabla \Psi$. We also find some of the Figure 3-5 too small and illegible, and question what the value is of including all these miniature box plots. Suggestions would be for these errors to be fixed, and to summarize these plots more succinctly in the main text.

Response:

We apologize for the errors and thank you for catching them. We have checked and corrected the equations. We have also summarized some of the less consequential boxplots in the main text and moved them to the supplement. For Fig. 2d, we have corrected the y-axes; the first y-axis is labeled with ψ which represents potential, the second y-axis is $-\nabla \psi$, which represents the drift. We have also corrected the notation mistakes in the methods section. Finally, we have condensed the boxplots in Fig. 5 and moved larger panels of perturbations to the supplement to enhance clarity.

Reviewer #2 (Expertise: Analysis of time-resolved scRNASeq data):

Yeo, Saksena and Gifford present a framework for modeling differentiation trajectories from single cell RNA-Seq data. Although there has been a lot of previous work in this area, this manuscript sets itself apart by modeling the process over real time rather than pseudotime and by being a generative model, thus allowing for interesting *in silico* perturbation experiments. I believe the main contribution of this manuscript is to extend the previous work of the authors (Hashimoto, Gifford and Jaakkola 2016), incorporating a model for cell proliferation as well as extensively demonstrating and benchmarking the framework of Hashimoto et al. and its applications to the single cell community.

Overall this is an important advance and a very welcome addition to the scRNA-seq analysis toolset with potential for wide use. However, the manuscript text needs some revision as it is currently impenetrable to a wider audience in my opinion. Furthermore, to ascertain the practical relevance of the method, the tool and code require documentation to make them usable by the wider community (<https://github.com/gifford-lab/prescient>), and some method and benchmark details need clarification.

Main Comments:

(1) I think the main practical attractive feature of this framework is the ability to simulate perturbations in time-course experiments, so I will focus here this main comment in this context:
- I wonder why the authors did not choose to perform a transcriptome-wide unbiased screen (they do a 200 TF screen). Is there a theoretical or practical limitation?

Response:

This is definitely a salient point, and, to answer your question, there is not a practical limitation in running a screen of every gene in the dataset (see below response on computational resources and runtime). We chose to concentrate on TFs since we cannot claim that our model can distinguish between causal and associative effects, and TFs are more likely to be causal. This is a limitation of using only scRNA-seq data for perturbational responses, as they can only be used as an associative tool for designing experiments.

- How much time and compute resources does the method require? Can the authors provide a computational benchmark, given certain GPU hardware how does the method perform vs cell number, and perturbation experiment number.

Response:

We agree that this would be a valuable piece of information for potential users of PRESCIENT. We have now included a computational resources section in the main Methods section and have added Supplementary Figures 1F and G (reproduced below):

F. Number of cells vs. run-time of training (gpu)

G. The number of simulations vs. training (gpu/cpu)

We would like to emphasize that, for a given longitudinal scRNA-seq dataset, training is one-time cost, and forward simulations (the costly portion of perturbational analyses) are computationally very inexpensive (akin to inference with a neural network classifier, for instance). Because of the nature of the training process (per cell simulations are mapped onto the GPU simultaneously), we recommend using a GPU framework for training models. For users without access to GPU hardware, PRESCIENT models can now easily be trained on AWS/Gcloud servers using their standard command-line interface provided at <https://cgs.csail.mit.edu/prescient>.

- There are recent attempts addressing similar problems (Lotfollahi et al. Nat Methods 2020), this is mentioned only in passing but further delineation of the differences would be very welcome. Also WOT, which the authors compare to in Figure 2 can provide some predictions of which TFs are important for cell fate - it's not exactly perturbation and so not directly comparable but still appropriate to include WOT results also in one of the TF case studies such as the one in Figure 5 (<https://nbviewer.jupyter.org/github/broadinstitute/wot/blob/master/notebooks/Notebook-8-predictive-TFs.ipynb>).

Response:

We have now included a discussion in the introduction as well as a figure (Figure 1a) of existing pseudo-time methods in the introduction and how PRESCIENT is different in that it operates in real-time (rather than pseudotime) and is a queryable, parametric model. Pseudotime methods only order cells, don't predict distributions over cell fates, don't predict perturbations, and are difficult to compare with lineage tracing methods because they are not generative, hence they don't produce predictions of cell states over time for comparison. "Model-free" or "coupling" models (like WOT and FateID) are limited in utility across datasets and for generalizing differentiation trajectories. For example, these models rely on observed training points and can not accommodate out-of-sample gene expression profiles, limiting the ability to predict the trajectories of perturbed cells. scGen shows the effect of a perturbation on an entire dataset in gene expression space. PRESCIENT shows the effect of a perturbation of a specific cell, at a specific time point over an entire differentiation process. scGen does not model real time or distributions of the perturbational effects i.e. given a single cell and the perturbation applied to it, it doesn't give a distribution over the perturbed cell, simply a deterministic shift in the latent space corresponding to a shift in the original space. It is not clear scGen could be simply applied for this use, and their original paper does not claim this function. scGen could benefit PRESCIENT models, as perturbed gene expression profiles could be more appropriate for initialization of PRESCIENT in perturbation mode but outside the scope of the study.

- The authors perturb the genes in the space of z-scores but how does the expression level of the genes affect the ability to predict their perturbation effects? Furthermore, the entire model is run in PCA cell space so changes due to specific genes are difficult to disentangle. For example, only the top 2500 variable genes are used to derive the principal components in one part of the study.

Response:

Great question, we now include a scatter plot of average normalized expression of each perturbed TF at the final timepoint vs. change in cell fraction of simulated cells (Figure S4d, reproduced below). This plot shows that there is not a correlation between average expression and the outcome of perturbations to the model. Like you mentioned, we use the highly variable genes (as identified by standard pipelines like Seurat and Scanpy) to build the PCs for analysis. This guarantees that the genes included are expressed and relevant to cell type specification. We agree that using PCA space is not always the ideal approach, partially for some of the reasons you mentioned. However, because PCA is a linear operation, the distances in PCA space are more stable and relevant than other spaces, even gene expression space. The difficulty of

operating on gene expression space itself is also observed by other comparable models like WOT and scGen, which operate on PCA space and the latent space of an autoencoder derived from the top N highly variable genes, respectively. Many typical scRNA-seq downstream analyses like clustering rely on PC space, as well. We want to clarify that we are interested in consequences of perturbations on cell *state* and not on the expression of specific genes.

Figure S4d. Normalized gene expression at timepoint 0 vs. Log2(fold-change) in cell fractions from perturbations of 200+ TFs ($z=5$) for β -cells (purple), α -cells (red) and EC-cells (blue)

- There is a focus on TFs, I guess the motivation is that TFs define the waddington differentiation landscape in principle, but from a performance point of view: what happens if essential genes are perturbed? what happens if signaling molecules are perturbed? Is it appropriate to use this model to study these perturbations even though paracrine effects are not accounted for by the model?

Response:

In the supplement, we show no effect for random gene sets that are not part of proliferative signatures. As we mentioned earlier, we are wary of perturbing non-TFs because of the causal vs. associative claims we could make. We are definitely interested in modeling perturbational effects beyond transcription factors, and we believe perturbing “gene modules” like single molecule perturbations of signaling pathways could be of great use to biologists. This would constitute a large advance of the model, especially with respect to generating proper perturbational profiles of various perturbagens to initialize simulations, and we are continuing to work on this for future versions of PRESCIENT. Approaching the perturbations of non-TFs in this way ensures we can maintain some level of causality in outcomes of the model.

- In figure 4c, what is the observed expression of the TFs at the evaluated time-points? Also in Figures 5a,b.

Response:

We have added a figure in the supplement to show expression for select endocrine induction TFs over the provided timepoints (Fig. S4b, reproduced below).

Fig S4b. Normalized expression of select TFs over time-points

(2) The main code repository of this work is provided here: <https://github.com/gifford-lab/prescient>, however without any documentation or code comments. This is not about reproducibility of this paper's results (although that's also important) but about whether others can easily test and use the methods for their own data. I strongly encourage the authors to invest the time in providing accessible tutorials and documentation that clearly outline input file formats, expected outputs, what the different functions do and so on. The authors can take the tutorials provided by popular scRNA-seq tools such as Seurat and Scanpy as examples. It's worth the effort because it will increase the impact of this work significantly and would've helped clarify many of the questions below.

Response:

We agree that the original codebase was difficult to use, and we have made the following adjustments:

- We packaged PRESCIENT as a PyPI package (<https://github.com/gifford-lab/prescient>) that can be easily installed and run.
- We now have a website <https://cgs.csail.mit.edu/prescient/> with complete command line documentation, quickstart, analysis vignettes, and input formats.
- We refactored the codebase to make the scripts dataset agnostic and specified a standardized input format for training models and running perturbation simulations with already trained models.
- We have made a separate repository (<https://github.com/gifford-lab/prescient-analysis>) containing notebooks to reproduce the figures presented throughout the paper. In addition, all training data, trained models, perturbational outcomes in the paper have been uploaded to a Dropbox (see paper highlights for links).

We have also provided this information in the code availability section.

(3) In many places in the text, previous specialized knowledge related to specific papers is assumed. This can be improved to make the paper accessible. I would recommend the authors go through and make sure all important concepts are explained in the main text when possible and further elaborated in supplementary notes. I quote a few examples that the authors can choose to revise:

- lines 45,46: please elaborate what the other methods do - what does "modeling capacity" mean? This is important to elaborate in the main text because it's related to how this work is placed within the context of other similar methods. Supp. Note 1 might be good to include in the main text if possible.

- line 49: What is the model in Hashimoto et al. and how exactly does this paper extend it? (one starts to infer this only if we read the supp. methods or Hashimoto et al. The latter nicely explains the motivation but this needs to be elaborated also here.)

- line 53: "cell fate bias". this phrase is repeated often but not clear to me what it means at this stage in the text and whether its meaning slightly changes in different places.

- Section starting at line 118: it wasn't elaborated how cell proliferation is incorporated. perhaps a brief explanation and reference to the supplemental methods is appropriate. Please also see comment (8)

Response:

Thank you for bringing this to our attention, and we apologize for the lack of clarity.

(Lines 45-46): We have now included Supplementary Note 1.1 in the main text and further elaborated on how PRESCIENT is different from existing methods

(Line 49): We have now included a description of the contributions of Hashimoto et al's work, and how our work extends his model

(Line 53): We have now cell fate bias when it is first introduced in the manuscript

(Section starting at line 118): We have now included further intuition for how cell proliferation was incorporated into the model.

Other comments & questions:

(4) Using correlation as a benchmark is not very convincing (example: figure 2b) - the impressive p-value is likely because there is a lot of data points, not sure this is really indicative of a correlation. Also please double check Figure 2f, and please provide the scatter plots in the supplement.

Response:

We agree about the limitations of Figure 2b, and while our procedure for estimating growth rates is imperfect, we observed that including these estimated growth rates consistently improves our model performance. In addition, this method has been previously used in WOT (Schiebinger, et al.). We report the correlation to be transparent in how we derived growth rates, and the fact is that the empirical growth rates from lineage tracing are also noisy. We have also amended the text to refer to the lineage tracing derived growth rates from the Weinreb et al. data are *empirical* growth rates, rather than *ground-truth*, since we cannot assume the lineage tracing data is perfect. We agree there needs to be general improvements in how the community computes proliferation

rates, especially as cell fate simulations become more and more possible, but this is outside the scope of this paper.

For correlation metrics used to evaluate cell fate bias, we provide this because it was previously used by Weinreb et al. to measure concordance with experimental lineage tracing. We note that the number of cells computed for the correlation in figure 2f is a lot fewer ($n = 335$ in the test set, see figure 2e reproduced below) and is the same test set used in Weinreb et al. 2020. Furthermore, we plot the empirical and predicted fate bias distributions in figure S2b to visualize their concordance. However, as mentioned in the manuscript, we observed the measured fate bias to be highly bimodal, and hence also provide AUROC of classifying a given cell as having a clonal fate bias of > 0.5 , which may be a better metric for performance.

Fig 2e, Summary of training/test splits of lineage tracing dataset. Training splits either include only cells with lineage tracing data, all cells, or cells without lineage tracing data.

(5) Wasserstein distance is used during model fitting as the main constraint (minimize the Wasserstein distance between observed data and the model fit, line 521) - but also sometimes as a benchmark (example: Figure 2a). Use as a benchmark requires explaining what it is in the main text and justification why it's a good way to measure performance relative to the model of WOT.

Response:

We have added a better description of the interpolation task in the main text, but we will also describe our experimental setup here. Wasserstein distance -- which is an extension of Euclidean distance in the population setting -- is minimized in our objective function to train the drift parameter neural network, as we minimize this distance with observed scRNA-seq time points to update the neural network that parametrizes the drift function of the diffusion process. When we use Wasserstein distance in the Supplementary Figure 2, we are evaluating this on *held-out* timepoints that are not used in training, and a lower value indicates that we were able to successfully interpolate between time point dynamics without observing. For example, in one task we use day 2 and day 6 to train a PRESCIENT model and then evaluate on day 4. This type of held-out analysis with an analogous metric (MSE) is common in machine learning applications.

Finally, we choose to include this interpolation task, as it was one of the only forms of validation used in Schiebinger et al when testing WOT, so we deemed it important to include a comparison. We chose to move this to supplement because we don't believe it is the most relevant display of utility for generative models like PRESCIENT, the cell fate prediction and perturbation tasks are more representative of methodological advances. Further, we refer you to Schiebinger et al. 2019's justification for geodesic interpolation (held-out analysis): <https://www.cell.com/cms/10.1016/j.cell.2019.01.006/attachment/d251b88d-a356-436a-a9a7-b7e04b56b41f/mmc8>.

(6) The data is always represented as a UMAP, and the potential function and drift is represented on top of that. Can the authors also evaluate the cell pseudotime using any of the other methods on some of the data and present this, and also plot the cell physical time? this is helpful as it may give an intuition to potential performance difference in practice between the typical approaches and what the authors propose. Also I'm trying to understand if there is a more direct way to represent the potential function, what does the drift direction (arrows) mean in the independently-derived UMAP space? Would be very helpful if this is clarified explicitly in the manuscript.

Response:

Thanks for pointing out that this is unclear. We have now added a description of how the visualization is generated and some interpretation for the figure's meaning in the methods section. We use UMAP as it is the standard approach for visualizing scRNA-seq data currently, and is convenient for representing this data in 2-dimensions, but other visualizations may be used. We have also added a drift visualization in two PC space to illustrate that the projection is arbitrary but the information is derived from the potential landscape learned on 50-PC space.

(7) I wonder how the model performance might be affected by the time sampling density. I imagine at some point, predictions would break as sampling density decreases depending on the complexity of the differentiation process in question. If the authors have evaluated this, it might be good to mention in the discussion since it will help with experiment design considerations if one is planning to use this method.

Response:

Thanks for the insightful observation, and, yes, we expect that the model improves with increased sampling density and have included that comment in the discussion. With the current longitudinal data available, this is difficult to show. Weinreb et. al. (2020) provides the first large dataset with lineage tracing + scRNA-seq time-series of a differentiation project, but only provides 3 time points, and Veres et. al. (2019) provides more dense time-points but no lineage tracing data. We do not have ground truth to compare with under different sampling densities. As more datasets with ground truth lineage tracing data and densely sampled time-series emerge, we intend to include this analysis in the documentation of the project for users. Hashimoto et. al (2016) showed that, in theory, the potential function can be recovered with only 2 time-points under certain assumptions, of which relevant to this discussion are 1) that the sampled points are cross-sectional snapshots of a dynamic process and 2) the final time-point is "well mixed". In reality, these assumptions may be broken to different degrees with scRNA-seq.

(8) I have to admit after reading the section starting at line 526 I still did not understand (ie. do not have a clear intuition) how exactly the proliferation rates would affect the objective function on line 521. As this is probably the main extension relative to Hashimoto et al and key to performance improvement (Figure 2), I would like to kindly ask the authors to elaborate this further. Also related to this, it was a bit surprising to me that basing proliferation on ground truth rates did worse than when based on KEGG gene sets (Figure 2f-g).

Response:

We apologize for the lack of clarity on how the proliferation rate affects the objective function of PRESCIENT's core inference task. We have updated the text to make this clearer, but we will also respond here. The objective function relies on minimizing the Wasserstein distance between simulated points from PRESCIENT's potential function and observed points when available. The Wasserstein distance calculation is itself an optimization problem in which the distance required to move a point in the source (observed cells) and target (predicted cells at that time step) is minimized for each point. The estimated proliferation rates are incorporated by weighting the source population according to the number of cells in the target population in the Wasserstein distance computation itself. The intuition behind this is that a cell that has a larger number of descendants should be mapped to a larger number of cells at the later time point. Across all cells, this provides a single number that is then minimized in our objective function during training. To address your second point, we also were surprised by the result that "ground truth" proliferation rates taken from the lineage tracing data itself in the Weinreb et al. dataset performed worse than KEGG-estimated growth rates. We feel it was misleading on our part to call the growth rates derived from lineage tracing "ground truth" and have opted to call them instead "empirical growth rates", as these themselves are estimates with a fair bit of noise due to the infancy and limitations of lineage tracing techniques.

(9) does the model allow for stationary cells (such as senescent cells that were assayed at $t=0$) or would those correspond to infinite potential? - thank you also for the clarification in the discussion if appropriate.

The model should allow for stationary cells if the population of senescent cells remains constant. The model has 2 distinct ways to deal with this: stationary cells have 0 cell cycle, so 0 proliferation, so will never move out, or the model assigns arbitrarily high potential because the cells themselves are stationary in the longitudinal scRNA-seq.

(10) in practice, sampling a time-course for single cell studies can result in batch effects. were these taken into account during benchmarking? do the authors recommend a certain method for deriving principal components in those cases as input to their model, such as MNN or Harmony?

Response:

We followed the pre-processing guidelines from the original study for both datasets, which did not report pronounced batch effects. For datasets that do report batch effects, integration methods like MNN or Harmony may need to be applied. That being said, this is a current limitation of interpreting PRESCIENT models after training, since the transformation that integration methods apply are difficult to interpret, limiting the ability to introduce sensible perturbations in the

integrated space. As more integration methods arise, we anticipate this will become a non-issue, but solving this is currently outside the scope of this paper.

(11) Figure 4d is mentioned after Figure 5

Thank you for catching this, we have updated the order of the figures to make more sense with the flow of the text.

Reviewers' Comments:

Reviewer #1:

Remarks to the Author:

The authors have addressed all of my comments. In particular, I am impressed with the python package and the associated documentation which I now believe can be held up as a very nice example of what should be provided in 2021.

Reviewer #2:

Remarks to the Author:

Thank you for documenting the code, the notebooks and for making the method available as an easy to install package. Also thanks for the deliberative answers to my questions.

There is one minor observation: there are some references to "ground truth" proliferation rates in the figures rather than "empirically derived" proliferation rates as the authors indicated in their response and the manuscript text.

I have no further comments.

REVIEWERS' COMMENTS

Reviewer #1 (Remarks to the Author):

The authors have addressed all of my comments. In particular, I am impressed with the python package and the associated documentation which I now believe can be held up as a very nice example of what should be provided in 2021.

We would like to thank Reviewer #1 for the suggestion to develop a useful package for others to use and the high praise for said package. We also thank them for the comments and questions, which helped us focus our Introduction and Discussion.

Reviewer #2 (Remarks to the Author):

Thank you for documenting the code, the notebooks and for making the method available as an easy to install package. Also thanks for the deliberative answers to my questions.

There is one minor observation: there are some references to "ground truth" proliferation rates in the figures rather than "empirically derived" proliferation rates as the authors indicated in their response and the manuscript text.

I have no further comments.

We would like to thank Reviewer #2 for the very insightful comments and questions, which helped us focus our paper, contextualize results, and discuss future directions for the project. We have fixed the reference of "ground-truth" in Fig. 2, as shown below.